# Metabolic contrast agents produced from transported solid $^{13}$C-glucose hyperpolarized via dynamic nuclear polarization

Andrea Capozzi [1,2✉], Jan Kilund[2], Magnus Karlsson[2], Saket Patel[2], Arthur Cesar Pinon[2], François Vibert[3], Olivier Ouari[3], Mathilde H. Lerche[2] & Jan Henrik Ardenkjær-Larsen [2]

Magnetic Resonance Imaging combined with hyperpolarized $^{13}$C-labelled metabolic contrast agents produced via dissolution Dynamic Nuclear Polarization can, non-invasively and in real-time, report on tissue specific aberrant metabolism. However, hyperpolarization equipment is expensive, technically demanding and needs to be installed on-site for the end-user. In this work, we provide a robust methodology that allows remote production of the hyperpolarized $^{13}$C-labelled metabolic contrast agents. The methodology, built on photo-induced thermally labile radicals, allows solid sample extraction from the hyperpolarization equipment and several hours' lifetime of the $^{13}$C-labelled metabolic contrast agents at appropriate storage/transport conditions. Exemplified with [U-$^{13}$C, d$_7$]-D-glucose, we remotely produce hyperpolarized $^{13}$C-labelled metabolic contrast agents and generate above 10,000-fold liquid-state Magnetic Resonance signal enhancement at 9.4 T, keeping on-site only a simple dissolution device.

[1] LIFMET, Department of Physics, EPFL, Station 6 (Batiment CH), Lausanne, Switzerland. [2] HYPERMAG, Department of Health Technology, Technical University of Denmark, Lyngby, Denmark. [3] Institut de Chimie Radicalire Aix-Marseille Université, CNRS, ICR UMR 7273, Marseille, Cedex 20, France.
✉email: andrea.capozzi@epfl.ch

Changes in metabolic pathways, causal of the origin or progression of diseases such as cancer, diabetes, and neurodegenerative diseases, can be studied non-invasively using metabolic imaging methods[1–3]. These techniques represent powerful means to diagnose and monitor response to therapy[4]. Despite significant research progress, metabolic imaging is still far from optimal in patients and blind to many cellular processes[5].

Among those methods, [18]F-fluoro-deoxy-glucose positron emission tomography ([18]F-FDG PET) is the current benchmark to assess hypo- or hypermetabolism in clinical practice, in particular for what concerns cancer diagnosis, staging, and treatment monitoring[6]. Nevertheless, [18]F-FDG PET has its limitations. For instance, it suffers from poor specificity in organs with a high normal glucose uptake[7] and in inflamed normal tissue with the risk of false positives[8]. Moreover, the use of [18]F-FDG PET agents exposes the patient to gamma-rays, which restricts its use, e.g., in certain patient groups and for repeated examinations[8].

A complementary, direct, and specific way to track metabolism in vivo is to follow the fate of exogenous substrates by [13]C magnetic resonance spectroscopy (MRS) or spectroscopic imaging (MRSI). These techniques allow characterization of tumors by measuring downstream metabolism enabled by good spectral resolution of the different metabolites[9]. However, [13]C-MRS and MRSI widespread use in the clinic is limited by low signal-to-noise ratio (SNR). The MR signal is proportional to the nuclear spins' concentration and polarization (i.e., the net alignment of the nuclear spins ensembles in the direction of the applied magnetic field, the so-called $B_0$). Because of its gyromagnetic ration, [13]C sensitivity is a fourth compared to proton MRS, and its natural abundance is only 1%. Signal averaging is the typical workaround to overcome low SNR. Unfortunately, the price to pay is poor temporal resolution and thus direct access to the highly informative real-time metabolic flux[10].

Deuterium metabolic imaging, a novel, noninvasive method, combines deuterium MRSI with oral intake or intravenous infusion of 6,6-[2]H-labeled glucose to generate three-dimensional metabolic maps[11]. Although straightforward to implement, this technique is challenged by difficult signal disentanglement at clinical magnetic field strengths.

[13]C-MRS limitations has benefitted from developments in hyperpolarization technologies[12]. Using hyperpolarized agents, the low SNR drawback of [13]C-MRS can be circumvented[13], and metabolic activity can be imaged non-invasively and in real-time in humans[14,15].

Dissolution dynamic nuclear polarization (dDNP) is the most versatile among the methods to hyperpolarize small molecules in solution[16]. With dDNP, hyperpolarized (HP) metabolic contrast agents (MCAs) can be obtained with a 100,000-fold [13]C-nuclear spin polarization compared to thermal equilibrium on a clinical scanner (i.e., 1.5–3 T)[17]. The MCAs are produced in an expensive and technically demanding device known as dDNP polarizer. The polarizer provides the appropriate conditions of temperature and magnetic field to transfer polarization from unpaired electron spins, added to the sample in form of organic radicals, to [13]C nuclear spins borne on the MCA molecules, using microwaves irradiation. Shining microwaves at a frequency slightly higher or lower with respect to the electron spins resonance (ESR), polarization can be transferred from the electrons to the nuclei, thanks to their dipolar coupling.

Whereas, it typically takes hours to create a single injectable dose of MCA, the HP MCA's lifetime is only minutes after dissolution and extraction from the polarizer. Currently, to equip an MR facility with hyperpolarization, the dDNP device must be located in near proximity to the scanner. Although dDNP HP [13]C-MRS has the potential to revolutionize diagnostic radiology enabling precision medicine and personalized healthcare[14,18–20],

this limitation threatens its development into widespread clinical use.

The [13]C polarization's half-life within the MCAs is several orders of magnitude longer when kept frozen at cryogenic temperatures, even in presence of radicals in the sample. This allows, in principle, transportation of the MCAs far away from their production site[21]. Unfortunately, a dDNP sample cannot be extracted as a frozen solid without losing its hyperpolarization[16,22]. The reason lies in the same radicals that are added to the sample to allow the DNP process to take place inside the polarizer[23]. These radicals induce nuclear spins relaxation that becomes prohibitively fast in the solid state  and at low magneticfield[24]. These are the conditions experienced by an HP sample when pulled far away from the high field of the DNP machine[21,25,26].

Lifting the mandatory presence of technically demanding and costly hardware at individual clinical sites could be realized, instead, if hyperpolarized MCAs were produced at a central facility for subsequent storage and distribution to the sites of action. Such remote production of [13]C-labeled MCAs could be envisioned to be much like the way clinical examinations are performed with [18]F-FDG PET, where the tracer is delivered on-demand.

The two key challenges to address are to extract the MCA as a frozen solid from the polarizer, while preserving its hyperpolarization, and to prolong the [13]C HP signal lifetime as much as possible under-sample transport conditions. Three different concepts have been published for producing long-lasting HP samples for storage and transportation[25–27]. The critical point in these concepts is the absence or drastic reduction of paramagnetic relaxation during extraction from the polarizer of the solid-state sample.

The first approach, proposed by Hirsch et al.[25], does not use DNP to increase the polarization of the substrate of interest. Indeed, no paramagnetic agents are added to the MCA formulation, which is hyperpolarized by brute force (e.g., cooling down the sample to very low temperatures while keeping it at high magnetic field). Although an easy solution to get rid of detrimental paramagnetism, this method is very slow (i.e., it takes tens of hours to thermally polarize the sample when working at 2 K and 14 T) and the obtained liquid-state hyperpolarization is between two and three orders of magnitude lower with respect to DNP.

The second approach, introduced by Ji et al.[26], physically separates the radicals from the [13]C-MCA in the sample. DNP is performed on [1]H nuclei of a radical doped solvent that impregnates without dissolving the MCA. Then, [1]H -[1]H spin diffusion and cross-polarization transfer the high spin order from solvent protons to [13]C nuclei borne on the MCA molecule. This is an elegant idea for combining hyperpolarization via DNP to HP sample extraction and transportation. However, the distance between the [13]C nuclei and the radicals in the sample matrix makes the DNP process less efficient by a factor of three, at least[26,28]. Moreover, the radical-rich phase of the sample involves non-biocompatible solvents such as toluene and tetrahydrofuran.

The third approach hyperpolarizes the sample employing labile radicals originating from alpha keto acids UV-light irradiation. These radicals are stable at low temperature, where the DNP process takes place, but recombine into diamagnetic species above 190 K (i.e., the radical quenching temperature)[29–31]. Therefore, a radical-free HP solid sample can be obtained by heating the latter above the UV-radicals quenching temperature. Hyperpolarization of [13]C-MCAs using photo-induced non-persistent radicals has been optimized and shown to perform as good as DNP using stable radicals[29,31,32], while employing endogenous/biocompatible substances only.

In 2017, Capozzi et al.[27] demonstrated in a proof-of-principle study that labile radicals for dDNP, generated via UV-irradiation

of a sample containing a fraction of [1-$^{13}$C]pyruvic acid, could be quenched inside the polarizer by means of a "sample thermalization procedure", while retaining most of the hyperpolarization obtained via DNP in the solid state. A dramatic increase of the $^{13}$C $T_1$ was recorded after the removal of the radicals and this opened perspectives for solid sample storage. Unfortunately, lack of controlled sample extraction allowed no real attempt of transport.

In this work, we exploit UV-induced radicals to generate highly polarized $^{13}$C-MCAs in the liquid state with no need for dDNP polarizer on site. We demonstrate transportation at cryogenic temperature of such samples. We establish a robust protocol for sample loading, polarization, thermalization, extraction, transport, and dissolution away from the production site of the HP $^{13}$C-MCA solid sample. We tested our method on a single sample preparation involving deuterated trimethylpyruvic acid ($d_9$-TriPA) as UV-radical precursor and [U-$^{13}$C, $d_7$]-D-glucose as substrate[29], the latter being a molecule showing increasing interest in the hyperpolarization community thanks to the rich metabolic pathways it can give access to[33–36]. Moreover, the shorter liquid state $T_1$ of glucose compared to pyruvic acid after dissolution, would greatly benefit from quicker handling time prior to injection in vivo, by reducing as much as possible the distance between the scanner and the dissolution device. This can be far from trivial when dissolving the sample directly from the dDNP polarizer.

Specific hardware development allowed us to solve two stringent physics problems: (1) efficient heating of the sample while retaining most of the polarization; (2) avoid fast spin-lattice relaxation at low-field (10 mT range) present even in absence of radicals. We implemented a custom-designed fluid path (CFP) with the purpose of diagnosing and solving experimental challenges as well as making it possible to run all steps of the experiment in a user-friendly closed system.

## Results

**CFP performance (sample loading, hyperpolarization, and radical quenching)**. The CFP (see Fig. 1, all technical details are reported in the "Methods" section) allowed us to investigate, in a robust and reproducible way, all steps involved in a "remote DNP" experiment employing UV-induced radicals. These included: (1) UV-irradiated sample loading into the dDNP polarizer while keeping it below the critical temperature of around 190 K, (2) hyperpolarization of the sample, UV-radicals elimination, (3) HP sample extraction from the polarizer, (4) HP sample storage/transport and, finally, (5) HP sample dissolution away from the production site.

After 5 min of UV-light irradiation $40 \pm 4$ mM ($n = 3$) of radical was generated into the solid sample (see "Methods" for details about sample preparation). After leak testing of the CFP (see Supplementary Fig. 1), the sample was loaded into the polarizer to perform DNP at 6.7 T/$1.20 \pm 0.05$ K. Under optimal microwave irradiation, $^{13}$C nuclei could reach a solid-state polarization of $45 \pm 5\%$ with a buildup time constant of $1300 \pm 10$ s ($n = 3$), in good agreement with our former study[29]. The polarization step of the experiment is illustrated in Fig. 2A, and a typical DNP buildup curve of the sample is reported in the green portion of Fig. 2D.

The quenching procedure is captured in Fig. 2A–C showing a sketch of the sample conditions with and without radicals. Figure 2D (yellow and orange portions) accounts for the NMR signal during radical quenching. Before blowing He gas onto the sample for 20 s at 6 bar of pressure, best results were obtained by first switching OFF the microwaves, then lifting the vial 15 cm above the NMR coil, outside the liquid He bath, and leaving it

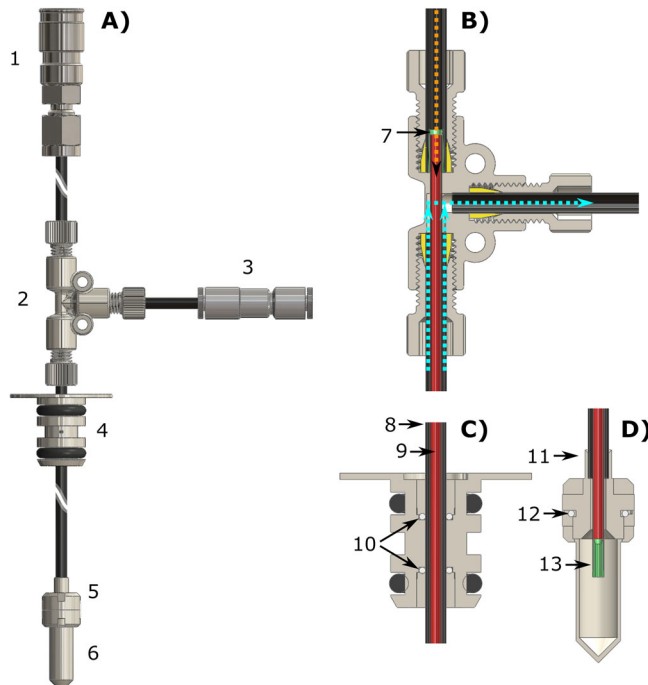

**Fig. 1 Custom fluid path (CFP) illustration.** Technical drawings of the CFP (**A**) and zoomed section of the T-valve indicating inner (orange arrow) and outer (cyan arrows) flow directions (**B**), dynamic sealing (**C**), and sample threaded vial (**D**). Numbers indicate the most important components of the device: quick release connection (1), T-valve (2), one-way valve (3), dynamic sealing (4), vial top part (5), vial bottom part (6), outer-lumen to inner-lumen transition (7), black PEEK outer-lumen (8), red PEEK inner-lumen (9), dynamic sealing silicon O-ring (10), laser-welded joint (11), vial PTFE O-ring (12), nozzle (13).

there for 5 min (see absence of recorded NMR signal in the yellow portion of Fig. 2D). The different parameters (i.e., blowing time, blowing pressure, quenching position, and waiting time) of the procedure were optimized in repeated experiments. This allowed us to get rid of 99% of the radical in the sample (see Supplementary Fig. 2), while measuring a polarization loss of 20% of the initial value (see orange portion of Fig. 2D), when reinserting the vial into the NMR coil (see Fig. 2C). The inset in Fig. 2D shows the "signature" of a successful thermalization experiment: the signal increased in the first few recorded NMR spectra.

Quenching of most of the radicals was confirmed by the absence of any DNP process when switching the microwaves back ON, and it caused a dramatic increase of $^{13}$C nuclear spin-lattice relaxation time. The latter, measured at 4.2 K and 6.7 T, increased from $2300 \pm 20$ s for a non-quenched sample, to $200,000 \pm 3600$ s (i.e., $55 \pm 1$ h) for a quenched sample (see Fig. 3), confirming that the UV-radicals represented the main source of relaxation.

In a separate series of experiments, by implementing a manual field cycling inside the polarizer, we also measured the $^{13}$C relaxation of a sample after UV-radicals quenching at 4.2 K and 1 T (see "Methods" for details about the field cycling implementation). In Supplementary Fig. 3, we report the results: by fitting a mono-exponential curve to the data, we found a $T_1$ of $4.0 \pm 0.5$ h ($R^2 = 0.97$).

**Radical free solid sample extraction**. Despite quenching the radicals prior to HP solid sample extraction reduced the polarization losses from 90 to 10%, when exposing it to a magnetic field as small as 40 mT, lower values of the field made the polarization to relax completely (see Fig. 4A).

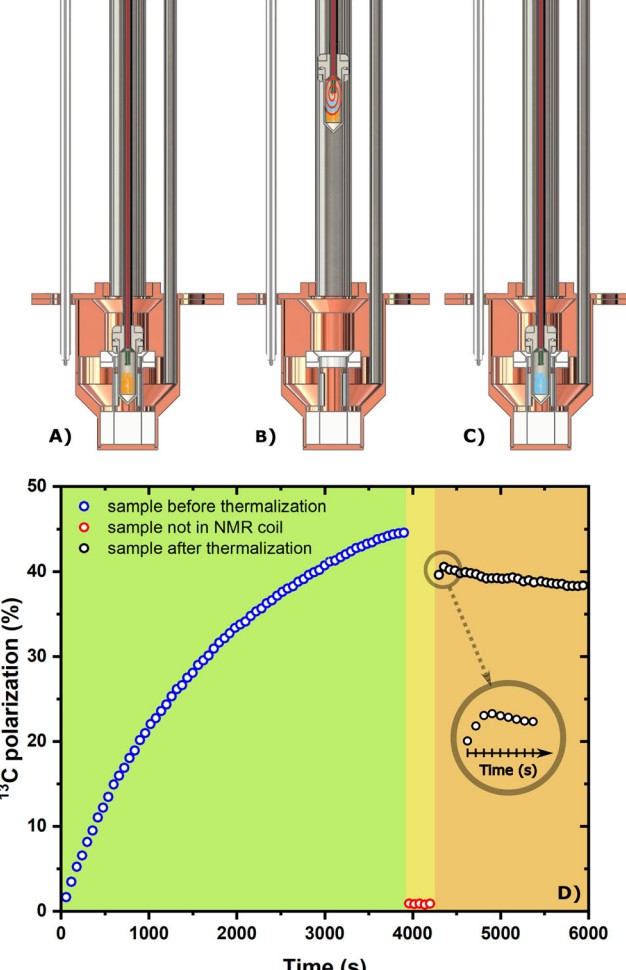

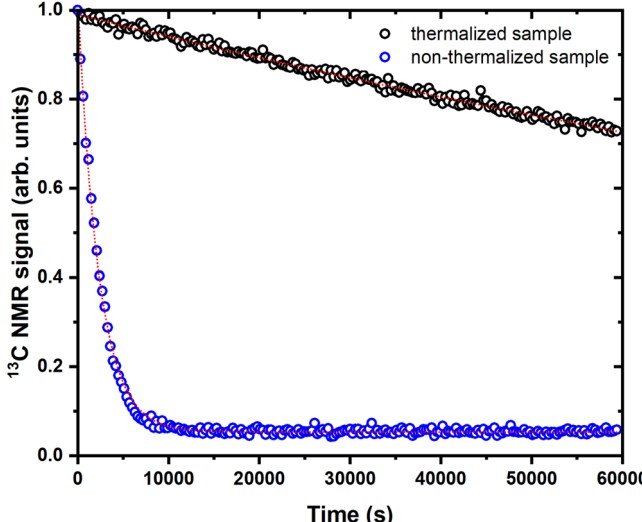

**Fig. 3 Extended spin-lattice relaxation for a sample after quenching of the radicals at 6.7 T and 4.2 K.** $^{13}C$ $T_1$ measurements for a thermalized (black circles) and non-thermalized sample (blue circles). Each experimental data point was acquired every 10 min. Red dotted curves are the result of a mono-exponential fit to the data. The $T_1$ values for the non-thermalized and thermalize sample were 2300 ± 20 s and 200,000 ± 3600 s, respectively ($R^2 = 0.99$).

**Fig. 2 Different steps of a radical scavenging experiment at 6.7 T and 1.2 K.** First, the sample sits inside the NMR coil and it is fully polarized. **A** A mono-exponential fit to the data provided a build-up time constant of 1300 ± 10 s ($R^2 = 0.99$, not shown). Second, the sample vial is lifted by 15 cm above the liquid He level and left there for 5 min; the CFP inlet is connected to a He gas line; room temperature He gas is blown on the frozen beads (**B**). Finally, after thermalization of the sample, the vial is lowered back inside the NMR coil for measurements (**C**). Yellow pellets represent sample beads before radical removal, while blue pellets represent sample beads after the thermalization process. The orange spiral represents the He gas flow inside the vial during thermalization. The NMR signal corresponding to panels (**A**), (**B**), and (**C**) is reported in panel (**D**) in the green, yellow, and orange portion of the graph, respectively; each data point was acquired every 60 s. The inset is a magnification of the first 90 s just after the sample goes back to measurement position. Each point was acquired every 10 s.

As the above results indicate severe relaxation due to exposure of the sample to a magnetic field lower than 40 mT, we modified the original DNP probe[37] by adding a "permanent magnets rail" providing a magnetic field of least 100 mT and oriented perpendicularly to the polarizer B₀ (see Fig. 4B). Details about the magnetic rail construction and magnetic field simulation are reported in the "Methods" section and in Supplementary Figs. 4, 5, and 6, respectively.

Repeating the extraction experiment with the magnetic enforced DNP probe, we were able to move a quenched sample from the polarizer isocenter to the loading chamber while retaining more than 90% of the polarization (see Fig. 4C).

It is important to notice that placing permanent magnets inside the DNP probe had no detrimental effects neither on the

homogeneity or shift of the NMR resonance nor on the polarizer base temperature, despite potentially increased heat conductivity.

**Sample transport and remote dissolution.** From field cycling experiments inside the polarizer, it was clear that hours long $T_1$ could be obtained for [U-$^{13}$C, d₇]-D-glucose at 1 T and liquid helium temperatures (see above). Since storage in liquid helium requires construction of a cryostat, we obtained the first results at liquid nitrogen temperature in a field of 1 T employing a simple transportation device (see Fig. 5A–E and "Methods" for details about the construction of the transportation device).

Once the sample vial reached the loading chamber, its disconnection from the polarizer and docking to the transportation device, lowering the sample into liquid nitrogen inside the storage magnet and reaching an NMR spectrometer placed 50 m far away from the polarizer took ~3 min. Once close to the NMR spectrometer, on-site dissolution generated a glucose polarization of 4.0 ± 1.0% ($n = 4$). One last optimization, aiming at speeding up the loading chamber disconnection, concerned the replacement of its vacuum clamp with a quick release one. This improved the measured $^{13}$C liquid-state polarization to 9% ($n = 1$) (results reported in Fig. 5F). We encourage the reader to watch the video recorded about the hyperpolarization transport and remote dissolution (see Supplementary Movie 1).

## Discussion

In this study, we exploited UV-induced labile radicals for DNP and unconventional hardware design to demonstrate, for the first-time, transport at cryogenic conditions and remote dissolution of HP [U-$^{13}$C, d₇]-D-glucose. Implementation of permanent magnets inside the DNP probe was a crucial step needed to successfully shelter the hyperpolarization during sample extraction. In vision of a distribution of HP MCAs on a larger scale, we envisage that several CFPs could be prepared by trained personnel and delivered on-demand.

At this stage of the study, a careful evaluation of the polarization losses is important to suggest further improvement of this

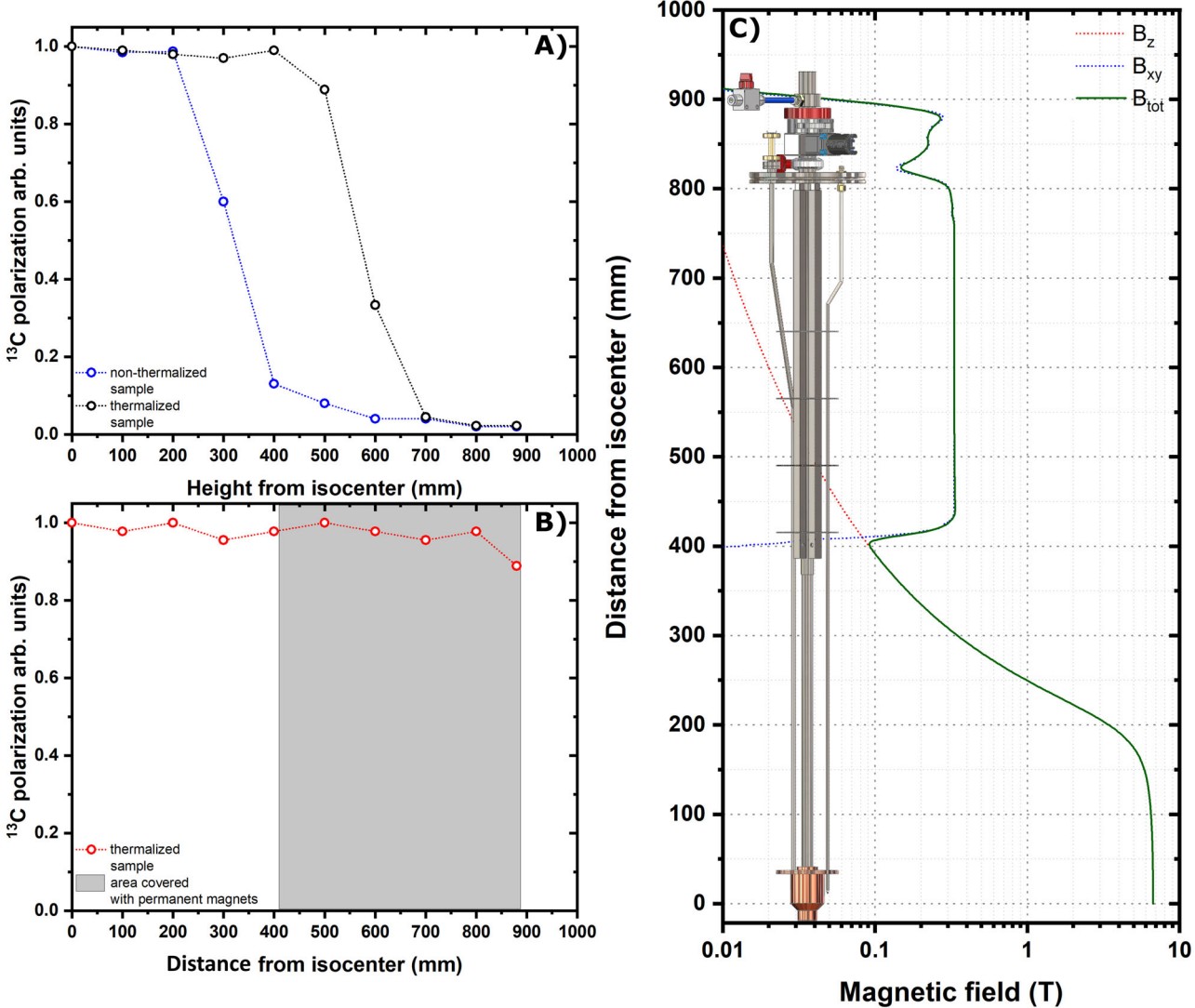

**Fig. 4 Sample extraction and hyperpolarization sheltering method.** $^{13}$C polarization losses as a function of the sample vertical position inside the polarizer while using a traditional DNP probe. The experiment was repeated for a non-thermalized sample (blue circles) and a thermalized sample (black circles) (**A**). $^{13}$C polarization losses as a function of the sample vertical position inside the polarizer while using the DNP probe equipped with permanent magnets. The experiment was performed for a thermalized sample only (red circles). The gray shaded area represents the area covered by permanent magnets in the probe. The last point was measured after lifting the sample up to the loading chamber and closing the mini-gate valve for 10 s (**B**). Calculated magnetic field value as a function of the distance from the polarizer isocenter. The magnetic field generated by the polarizer coil is parallel to the probe axis (red dotted line), while the magnetic field generated by the permanent magnets is perpendicular to the probe axis (blue dotted line). The norm of the total field is also reported (green continuous line) (**C**).

technique as a game changer for production of transportable DNP hyperpolarized MCAs.

As earlier reported[29], a $^{13}$C liquid-state glucose polarization of ~30% is obtained when the sample formulation used in this study is dissolved directly from the dDNP polarizer (10 s interval between dissolution onset and start of the NMR acquisition on the 9.4 T magnet, with a reported [U-$^{13}$C, d$_7$]-D-glucose T$_1$ in solution of 20 s). When compared to our best result, we still lose 2/3 of the polarization during the extraction and transport process.

We characterized one source of relaxation in the experiment. The UV-radicals quenching process accounts for a relative polarization loss of 20%. This would project the maximum achievable liquid-state $^{13}$C polarization for glucose to 24%. According to the data reported in Fig. 4B, lifting a UV-radical quenched sample to the loading chamber causes almost no loss of polarization.

Moreover, if the gate valve was opened and the sample subjected to the cold He gas stream, performing a fast extraction (10 s) compared to a slow one (~2 min) did not make any difference.

A more potent source of polarization loss is the unavoidable relaxation of HP glucose when kept at 77 K and 1 T for the time needed to reach the site where the dissolution took place. Unfortunately, we did not find in the literature T$_1$ data for glucose at such experimental conditions and, for the time being we do not dispose of the needed hardware to perform these measurements. We can, however, provide a rough estimation of the T$_1$ at these transport conditions (1 T, 77 K) by referring to data published by Hirsch et al.[38] for solid [1-$^{13}$C]pyruvic acid, at 1.3 T and 60 K, with no radicals present in the sample. Under these conditions, the T$_1$ measured over 5 min for a partially annealed pyruvic acid sample and remained constant down to 20 K[38], where the methyl groups rotation is supposed to be minimal[39]. Assuming a similar value of T$_1$ for our sample at 1 T and 77 K (i.e., 5 min), this would

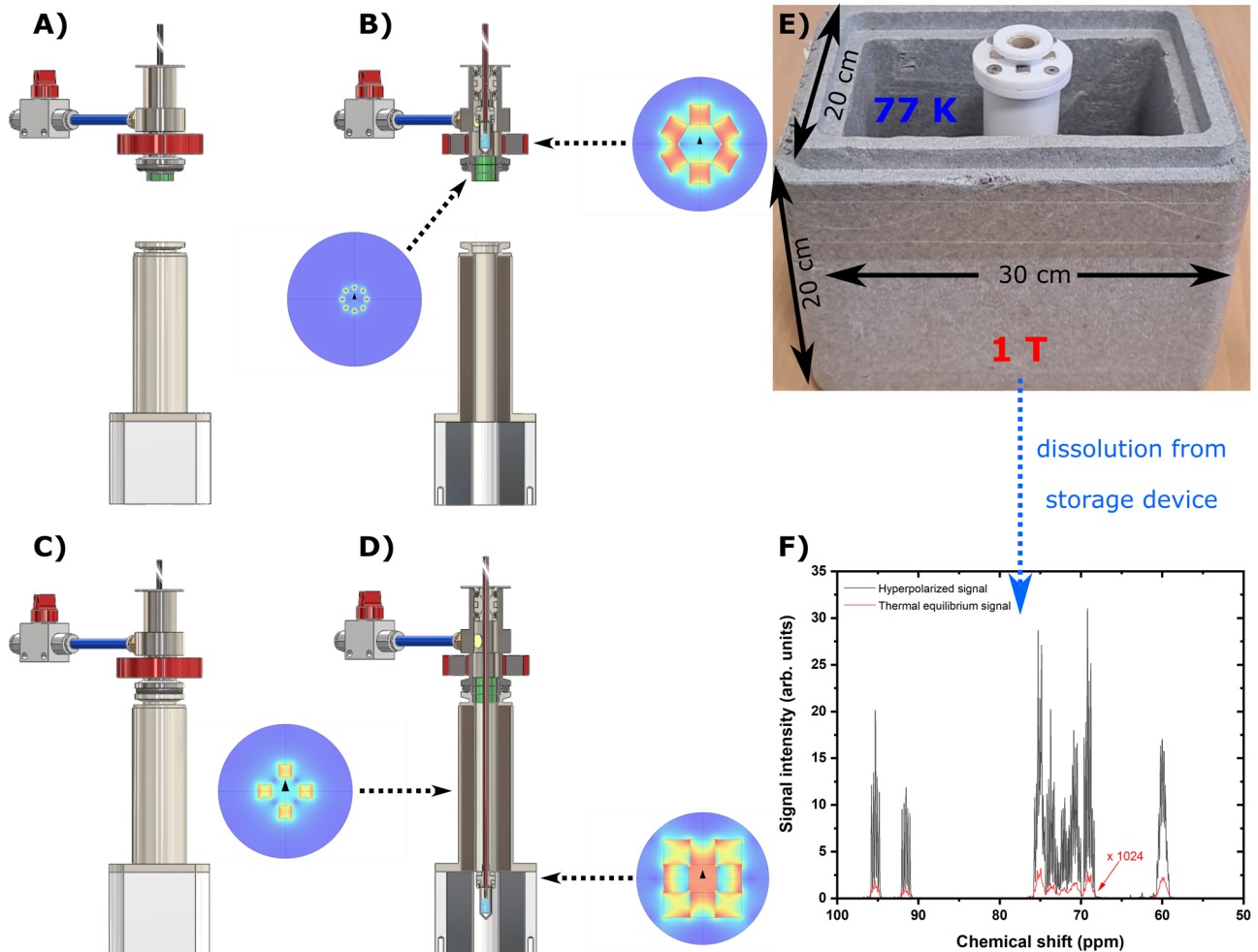

**Fig. 5 HP solid transport and remote dissolution.** We report here a schematic of our strategy for storage/transport of a HP sample and remote dissolution. After lifting up the sample vial above the gate valve, the transport procedure entails four main steps. First, the loading chamber/air lock is disconnected from the polarizer and docked to the transport device (**A**) and (**B**). Second, the sample vial is pushed down to reach the 1 T magnet (**C**) and (**D**). The transport device is composed by two parts: a four elements 300 mT Halbach array magnetic guide at the top and an eight elements 1 T Halbach array storage magnet at the bottom. The transportation device is placed inside a Styrofoam box and plunged in liquid nitrogen (**E**). Third, the Styrofoam box is transported to the site where the dissolution experiment is going to happen. Fourth, the CFP quick release is connected to the dissolution station (see Supporting Material) and the HP solution obtained and transferred to the NMR 9.4 T spectrometer for measurements (**F**).

account for a relative polarization loss of ~50% during transport, projecting the maximum achievable liquid-state $^{13}C$ polarization value for glucose to 12–13%.

A third possible source of polarization loss could be the sample heating during the time interval needed to dock the loading chamber to the transportation device, when the He gas stream does not cool the sample anymore. The importance of this loss is supported by the improvement obtained by reducing the time to complete this operation by implementation of a quick-release vacuum clamp.

To provide conditions for longer storage and/or transport, a colder environment is needed (i.e., below 4.2 K). At liquid He temperature, hours-long $T_1$ can be obtained on $^{13}C$-labeled small molecules, as previously demonstrated[38] and confirmed by our relaxation measurements of a thermalized [U-$^{13}C$, d$_7$]-D-glucose sample, where a $T_1$ of 4 h was obtained at 1 T and 4.2 K. Indeed, we are currently working on a more advanced transportable small bath cryostat able to work both at 4.2 and 77 K and equipped with a 1 T Halbach magnet sufficiently homogenous to perform NMR on the HP sample extracted from the polarizer. This will allow a better estimation as well as reduction of the polarization losses.

In the present study, an interesting observation concerns the NMR signal profile just after the quenching procedure. As it is shown in the inset of Fig. 2D, a good quenching procedure was characterized by an increase of the signal just after plunging back the sample into the liquid helium bath. Being the $^{13}C$ several order of magnitude above the thermal polarization level, we ascribe this phenomenon to a temporary increase and subsequent stabilization of the NMR coil Q-factor. Indeed, just after the quenching procedure, the sample vial was "hot" and its insertion into liquid helium generated a pressure increase into the VTI up to 20 mbar. This local heating close to the NMR coil might decreases its Q-factor for the concerned time interval, resulting in a slightly smaller detected NMR signal at the beginning of the acquisition.

Conversely, when the He gas did not bring enough heat to the sample to quench the radicals, the effect described above was covered by a stronger initial decrease of the signal followed by a plateau. This suggests that, in a Thermal Mixing interpretation of the phenomenon, the losses during the thermalization process were due to thermal contact between the $^{13}C$ nuclear spins reservoir and the $^{1}H$ nuclear spins reservoir mediated by the radicals non-Zeeman reservoir[40–42]. Indeed, when heating up the

sample, the protons relaxed very quickly due to efficient spin diffusion. If radicals are present in the sample, energy exchange may be maintained between the different nuclear spin pools. In this case, [1]H nuclei will "drain" polarization from [13]C nuclei until the same spin temperature is achieved. Given that this interpretation is correct, deuteration of the full sample matrix might help in reducing polarization losses during the quenching procedure. Loss of polarization due to heating of the sample rather than to a weaker magnetic field was supported by the fact that the sample vial could be kept for 5 min at 15 cm above the polarizer isocenter with no loss of polarization. This step was also useful to gently increase the sample temperature and therefore facilitate the radical quenching procedure. Moreover, during this process we chose to switch off the microwaves to avoid any undesirable nuclear spin relaxation effect caused by microwaves propagation outside the cavity[37].

It is worth noting that, with surprising reproducibility, 1% (i.e., 0.4 mM) of the initial radical concentration was surviving the quenching process. This small amount of radicals did not generate any particular problem during sample extraction from the polarizer when, as in this study, the magnetic field is sufficiently high[21]. This observation, together with the evidence of no radical leftover in the liquid state[29,30], suggests that the radical quenches in two steps: the majority at 190 K and the leftover at higher temperature.

Also, we want to stress that even in the ideal case of complete absence of paramagnetic impurities in the sample, for a magnetic field value <40 mT [13]C and [1]H nuclei are subjected to "low-field thermal mixing"[21,26]. When the Zeeman splitting difference between [1]H and [13]C nuclear spins is small compared to the width of the two resonance lines, energy can be exchanged between protons and carbons. Thus, if the [1]H spins' order is poor, due to relaxation during the thermalization process, polarization will be drained from [13]C spins until the two reservoirs achieve the same spin temperature. From the technical point of view, choosing permanent magnets that generate a magnetic field perpendicular to the polarizer's $B_0$ allowed us to avoid any zero field. For instance, addition of cylindrical permanent magnets with magnetization along the $z$-axis would have been less challenging, but such permanent magnets are characterized by an inversion of the direction of the magnetic field at the edges of the cylinder.

Finally, we want to draw the attention of the reader to the fact that our methodology can pave the way toward transportation of HP MCAs in the solid state, dispensing from the presence of a DNP polarizer at individual clinical sites. After transportation, the HP sample still needs to be dissolved. Because of the compact dimension of the transport/storage device, this operation could be performed on the side of the MRI scanner reducing the handling time of the HP solution. Nevertheless, all dDNP limitations related to the MCAs' relaxation time in solution still stand.

Our next aim is to broaden the applicability of our technique to other metabolic relevant molecules (e.g., pyruvate, urea, fumarate) and to start covering longer distances with the HP MCAs. By means of more advanced transportation devices, we plan to perform the hyperpolarization in our lab and run HP MRS animal experiments in a clinical environment.

## Methods

### Experimental design and implementation

*Custom designed fluid path—CFP.* The CFP was designed with the aim of being reusable, easily loading frozen solid samples and delivering to final users a "plug and play" closed device able to provide hyperpolarized injectable solutions with no need for a dDNP polarizer on site. The CFP is a combination of commercially available and custom-made components. Differently from the SPINlab version[43], it does not present any glued joints between the parts in order to improve durability and limit failures during dissolution. Moreover, all plastic parts are made from

polyether ether ketone (PEEK) or polyamide-imide (PAI). Refer to Fig. 1 for the following description.

Figure 1A reports the device in its entirety. From top to bottom we find: (component 1) a stainless steel quick release (SS-QM2-B-200, Swagelok, Solon, OH, USA) that can be connected to a buffer boiler/dissolution head for dissolution or a He gas line for sample thermalization; (component 2) a modified PEEK plastic T-valve (P-713, IDEX Health & Science, Lake Forest, IL, USA); (component 3) a one-way valve (AKH04-00, SMC, Tokyo, Japan); (component 4) dynamic sealing; (component 5) top and (component 6) bottom part of a PEEK threaded vial[31]. Figure 1B, C and D shows zoomed section views of the modified T-valve, dynamic sealing, and threaded vial, respectively. A PAI conical transition (Fig. 1B, component 7) connects the PEEK outer lumen (Fig. 1C, component 8) and inner lumen (Fig. 1C, component 9) inside the top arm of the T-valve and prevents any back flow toward the quick release. The T-valve's interior is modified to make a press fit between the inner lumen and the top arm of the T-valve while maximizing the flow in the bottom and right arms to split the gas/liquid inflow (orange arrow) from the outflow (cyan arrows). PEEK tubing is produced on demand (Zeus Inc., Orangeburg, NC, USA). The inner lumen (OD = 1.8 mm, ID = 1.6 mm) is extruded from natural PEEK (depicted in red in the figure to help distinguishing the different parts), while the outer lumen (OD = 3.2 mm, ID = 2.4 mm) is extruded from PEEK containing a black pigment. The black pigment is necessary to laser weld the top part of the vial to the outer lumen (Fig. 1D, component 11). The laser welding is performed by Leister Technologies (Kaegiswil, Switzerland). Compressing a PTFE O-ring (Fig. 1D, component 12) between the top and bottom part of the threaded vial yields a leak rate $<10^{-8}$ mbar·L/s at room temperature. The integrity of the laser welding and PTFE O-ring sealing was also verified by immersing the bottom part of the CFP in liquid nitrogen and pressurizing the device with He gas to 4 bar, without observing any pressure drop after 5 min. The inner lumen ends with a press fit PAI nozzle (Fig. 1D, component 13) to improve dissolution performance. The dynamic sealing allows to load and unload the dDNP sample inside the polarizer while keeping it constantly at low pressure. Leak tightness as good as $10^{-8}$ mbar·L/s is achieved by compressing silicon O-rings (Fig. 1D, components 10) around the outer lumen using purpose-made threaded plugs. It is important to notice that the T-valve is passive. Therefore, to avoid cryo-pumping during sample loading and polarization, the one-way valve remains always connected to the right-arm of the T-valve. The dissolution capability and reliability of the CFP were tested in a series of 10 consecutive experiments. The vial was filled with 100 µL of a solution of blue colorant dissolved in glycerol:water 1:1 (v/v); the CFP was loaded inside the polarizer and let in superfluid He for 1 h; the dissolution was performed by heating 6 mL of phosphate buffer. No chase He case was used to blow out the sample. The dissolution was successful 10 times out of 10 with complete melting of the sample; the volume of HP solution collected inside a Falcon tube was 4.0 ± 0.5 mL. In a separate experiment, the CFP was left inside the polarizer overnight and the next day dissolution was successful as well. See next section and ref. [44] for details about sample loading and dissolution procedure.

*Magnetic enforced DNP probe.* The second crucial hardware implementation dealt with a DNP probe that could shelter the sample hyperpolarization during extraction. Figure 6 shows drawings of the magnetic enforced DNP probe: top part front view in Fig. 6A, top part top view in Fig. 6B, bottom part front view in Fig. 6C, bottom part section view in Fig. 6D, top part section view in Fig. 6E. Refer to this figure for the following description.

Differently from Ji et al.[26], we decided to equip the probe with NdFeB permanent magnets (Supermagnete, Gottmadingen, Germany) along the path traveled by the sample instead of winding the sample vial inside a small solenoidal electro-magnet to simplify extraction operations.

To cover with permanent magnets the full path's length experienced by the sample during extraction, we designed four different Halbach array arrangements. A four elements Halbach array (7.5 mm × 7.5 mm × 100 mm bar magnets), held in place by stainless steel squared profiles (Fig. 6A, components 8), surrounds the probe stem (Fig. 6C, components 12). The last four elements of this "magnetic rail" enter the top flange by 90% of its thickness (see Fig. 6E, bottom inset). To fill the magnetic field gap between the ISO-KF-100 flange (Fig. 6A, component 5) and the mini gate-valve (Fig. 6A, component 3), four octagonal Halbach arrays (1.5 mm × 1.0 mm × 5 mm bar magnets) are stacked inside the two KF-16 half nipples (see Fig. 6E, component 18). The mini gate-valve volume is covered by gluing on it a two elements Halbach array (30 mm × 30 mm × 10 mm block magnets) (see Fig. 6E, right inset). A second stack of three octagonal Halbach arrays covers the gap between the mini gat-valve and loading chamber (Fig. 6E, component 17). Finally, a 3D printed hexagonal Halbach array (12 mm × 12 mm × 12 mm block magnets) is placed around the bottom part of the loading chamber (Fig. 6A, component 2) to insure sufficiently high magnetic field during transfer of the sample from the polarizer to the storage vessel. Details about the different Halbach arrays field profile are reported in Supporting Fig. 5.

### UV-sample preparation and handling

All chemicals were purchased from Sigma-Aldrich (Brøndby, Denmark) excepted for the radical precursor deuterated trimethylpyruvic acid (d9-TriPA) that was synthesized in house. We worked with a single kind of sample whose preparation was optimized in a former publication[29]. [U-[13]C, d7]-D-glucose was dissolved in 60 ± 3 µL of glycerol:water 1:1 (v/v) to

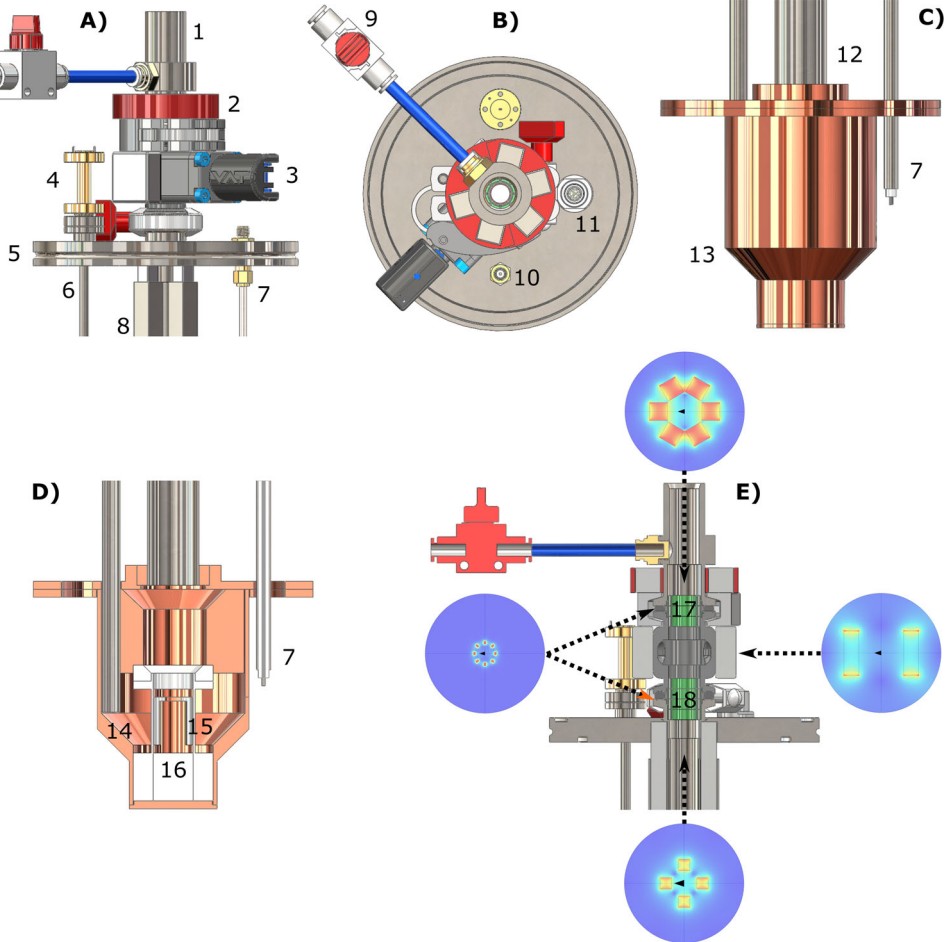

**Fig. 6 DNP probe for sample extraction.** Top and bottom part of the DNP probe are reported: top part front view (**A**), top part top view (**B**), bottom part front view (**C**), bottom part section view (**D**), top part front section view with simulated permanent magnets field profile cross-section (**E**). Numbers indicate the main components: sample loading chamber (1), loading chamber hexagonal Halbach array placed around loading chamber (2), mini gate-valve with NdFeB permanent magnets (3), WR5-to-circular microwave transition (4), ISO-KF-100 flange (5), circular waveguide (6), NMR rigid coax-cable (7), squared Halbach array around probe stem/loading tube (8), loading chamber He gas purging line (9), NMR bulkhead SMA connector (10), Fischer connector for liquid helium level meter (11), probe stem (12), microwave cavity (13), 45° microwave mirror (14), pseudo Alderman-Grant coil (15), PTFE coil former (16), octagonal Halbach arrays inside KF16 flanges (17, 18).

obtain a final glucose concentration of 2.2 M; d$_9$-TriPA was then added in amount corresponding to 10% of the final volume; the solution was sonicated at 40 °C for 5 min to efficiently degas the sample and improve the glass quality after freezing. A solution volume of 6.0 ± 0.3 μL was poured in liquid nitrogen as a drop to form a frozen bead. The operation was repeated 10 times. The frozen sample was transferred to a quartz Dewar (Magnettech, Berlin, Germany) filled with liquid nitrogen for UV irradiation. The irradiation set up was extensively described earlier[31]. UV-light was shined on the sample for 300 s using a broad-band source (Dymax BlueWave 75, Connecticut, USA) at full power (i.e., 19 W/cm$^2$). Refer to Fig. 7A for a simplified sketch of the setup. Radical concentration was measured immediately after irradiation by inserting the tail of the quartz Dewar in the cavity of an X-band spectrometer (Miniscope MS 5000, Magnettech, Berlin, Germany) and following methods described earlier[31]. Finally, the irradiated frozen beads were loaded inside the CFP vial bottom part (Fig. 7B, component 4). While keeping it and the bottom wrench (Fig. 7B, component 5) in a Styrofoam box filled with liquid nitrogen, frozen pellets were transferred, a new PTFE O-ring (Fig. 7B, component 3) put in place and squeezed by screwing the vial top part (Fig. 7B, component 2) by means of the top wrench (Fig. 7B, component 1). A leak test was then performed pressurizing the CFP with helium gas (see ref. [31] and former section for details).

Sample loading inside the polarizer proceeded as follows. The polarizer variable temperature insert (VTI) was kept filled with 10 cm of liquid He and at low pressure, the CFP was disconnected from the leak-test station (see Supporting Fig. 1) and the vial (Fig. 1A, components 5 and 6) quickly displaced from the Styrofoam box filled with liquid nitrogen to the loading chamber, while flushing the latter with He gas. The dynamic sealing (Fig. 1A, component 4) was lowered to close the loading chamber, the He gas flow shut down, and the gate-vale opened (Fig. 6A, component 3). The vial was manually pushed to the center of the NMR coil (Fig. 6D, component 15). The process took about 30 s in order to minimize

liquid He evaporation (VTI pressure <10 mbar). In its final position the vial touched the coil former (Fig. 6D, component 16).

**Microwave delivery and solid-state NMR measurements.** Once the sample was in place, microwaves were delivered from a 94 GHz solid-state source VCOM-10/94-WPT (ELVA-1, St. Petersburg, Russia) coupled to a 200×2R4 frequency doubler (VDI, Charlottesville, VA, USA), which provided an output power of 55 mW at 188 GHz. The source, digitally controlled through NI-DAQ device USB-6525 (National Instruments, Austin, TX, USA) has a tuning range of ±0.6 GHz and the possibility to modulate the output frequency at a rate up to 2 kHz and with an amplitude of up to 100 MHz.

Microwaves reach the probe cavity (Fig. 6C, component 13) traveling through a circular waveguide (Fig. 6A, component 6) ending with a 45° mirror (Fig. 6D, component 14) that reflects the microwaves toward the sample. In all experiments, microwave irradiation was performed at optimal conditions: the output power was 55 mW and the frequency was modulated, following a sinusoidal profile, at a rate of 1 kHz by 25 MHz around the central frequency 188.21 GHz. The latter corresponded to the negative maximum enhancement of the DNP spectrum[29].

All $^{13}$C NMR acquisitions were performed using a compact bench-top spectrometer (Kea2, Magritek, Wellington, New Zealand) connected to the DNP probe via a rigid coax-cable (Fig. 6, component 7). Details about the NMR and microwave delivery performances were published earlier[37,45]. The flip angle used for all acquisitions was 1° (pulse length = 5 μs; transmitted power = 5 W). The polarization build-up was monitored by pulsing every 60 s. Relaxation after thermalization was acquired by pulsing every 10 min. These measurements were performed at 4.2 K instead of 1.2 K because even in presence of radicals the relaxation time at 1.2 K can be several hours long, making it difficult to interpret

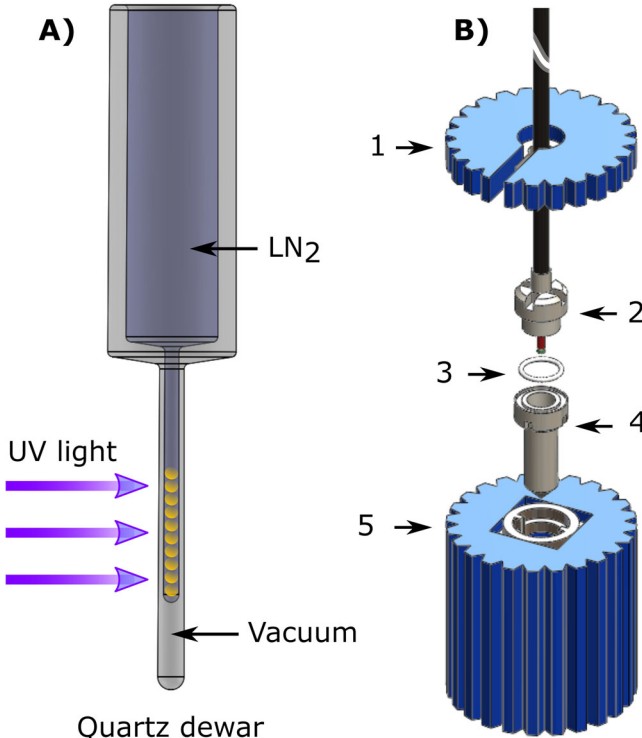

**A)**

LN$_2$

UV light

Vacuum

Quartz dewar

**B)**

**Fig. 7 UV-irradiation and sample vial loading.** The sample frozen beads are first UV-irradiated in liquid nitrogen (**A**). After irradiation and consequent radical generation, the sample is ready for DNP (**B**): the bottom part of the sample vial (4) sits inside the bottom wrench (5) in liquid nitrogen; the irradiated sample is transferred into the vial, a new PTFE O-ring (3) put in place and squeezed by the top part of the vial (2) using the top wrench (1).

the outcome of the quenching procedure. Differently, at 4.2 K amorphous solids enter a different relaxation regime (from direct process to Raman and Orbach)[46], and the $T_1$ becomes tens of minutes long when radicals are present. In absence of radicals, the $T_1$ increases to several hours at 4.2 K, making it straightforward to interpret the outcome of the quenching procedure.

The thermal equilibrium signal build-up was monitored overnight pulsing every 30 min, after saturation of any residual signal with a 50,000-rf pulses comb. The NMR signal was acquired every 30 min (1 average) until complete relaxation was achieved. The DNP enhancement was calculated by dividing the thermal equilibrium and DNP signal integrals.

**Investigation of the sample extraction path from the polarizer.** Thanks to the flexibility of the CFP dynamic sealing, the sample extraction path was recorded in steps of 10 cm from the magnet's isocenter to the loading chamber. For each step, the sample vial was lifted to the desired height, allowed to relax for 5 s, and then lowered back to the measurement position inside the NMR coil. The magnetic field profile of the superconductive magnet was measured with a Hall probe up to 3 T and simulated in MATLAB, according to the coil geometry, from 0 to 6.7 T (see Supporting Fig. 4). The simulation was in good agreement with the measured data points ($R^2 = 0.98$).

**Storage and transport of a radical free HP sample.** The transport and remote dissolution procedure is outlined in Fig. 5. The transport device was composed of two parts (see Fig. 5A–D): a 300 mT four elements Halbach array magnetic guide at the top and a 1 T eight elements Halbach array storage magnet at the bottom (see Supporting Fig. 6 for details). The transport device was precooled to 77 K by placing it into a Styrofoam box filled with liquid nitrogen (see Fig. 5E). The HP sample transport and remote dissolution entailed four main steps. Once the HP thermalized sample reached the loading chamber/air-lock, the polarizer's gate valve was closed, and the loading chamber/air-lock disconnected from the rest of the DNP probe. The loading chamber/air-lock was then docked to the transport device (Fig. 5A, B), and the sample was pushed down into liquid nitrogen to reach the storage magnet (Fig. 5C, D). The Styrofoam box was then put on a trolley, brought to a liquid state NMR laboratory installed two floors above the dDNP polarizer location, and the CFP connected to a compact dissolution station (see Supporting

Fig. 7) to extract the MCA in the liquid state. The HP MCA was finally injected into a 5 mm NMR tube and inserted in a 9.4 T vertical magnet to measure the polarization.

**Dissolution and liquid-state NMR measurements.** Six milliliters of 40 mM phosphate buffer containing 0.1 g/L of ethylenediaminetetraacetic acid (EDTA) was loaded into the CFP dissolution head/boiler (see Fig. S7) and pressurized to 4 bar with helium gas. The solution was heated to ~180 °C (12 bar of vapor pressure). For dissolutions from the polarizer, the VTI was kept at ~1 mbar, the CFP was lifted 15 cm out of the liquid helium, by sliding the outer lumen inside the dynamic sealing, and connected to an exit tube. The CFP inlet was then connected to the dissolution head, the hot buffer released, and the HP solution flushed out from the polarizer until the pressure in the dissolution head dropped to zero. The HP solution was recovered in a Falcon tube and manually injected into a 5 mm NMR tube. The superheated buffer reached the sample flowing through the CFP inner lumen. The melted sample came out from the polarizer flowing in between CFP inner and outer lumen. It finally flew through the one-way valve and reached the Falcon tube. When dissolving from the transportation device, all steps were as above, but the sample vial was not moved from the 1 T storage magnet and the dissolution happened while keeping the vial in liquid nitrogen.

All dissolved HP samples were transferred to a 9.4 T Varian (Palo Alto, California, USA) vertical high-resolution spectrometer for measurements. The decay of the $^{13}$C HP signal was monitored every 3 s with 5° pulses. Once complete relaxation was achieved the liquid sample in the NMR tube was doped with 10 μL of Dotarem® and reinserted into the spectrometer. The same 5° pulse was used to measure the signal corresponding to thermal equilibrium from 1024 averages with 1.1 s repetition time. The DNP enhancement was obtained computing the ratio between the value of the integral of the first spectrum of the HP decay and the thermal equilibrium one.

**Relaxation at longer-term storage conditions.** In a separate set of experiments, we measured the relaxation of a thermalized sample at 1 T and 4.2 K. To do so, we filled the polarizer with liquid He to a height of ~50 cm above the NMR coil and implemented a manual field cycling. A magnetic field of ~1 T corresponded to a vertical position of 25 cm above the isocenter of the polarizer (relaxation position). The field cycling happened as follows: the first data point was acquired after thermalization; the sample was then lifted to relaxation position and left there for 1 h; the sample was moved back inside the NMR coil to acquire the second data point. The procedure was repeated 5 times for a total duration of the experiment of 5 h.

**Other instruments, simulations, and data analysis.** The leak detector used in this study was a Phoenix from Leybold GmbH (Cologne, Germany). The Hall probe device was a Lake Shore 475 from Lake Shore Cryotronics (Westerville, OH, USA) equipped with a longitudinal and axial probe to measure the magnetic field from the superconductor and Halbach arrays, respectively. All NMR data were processed in MNOVA (Mestrelab Research, Santiago de Compostela, Spain). Magnetic field simulations were performed using MATLAB (Mathworks, Natick, MA, USA) and COMSOL 5.4 (COMSOL Multiphysics, Burlington, Massachusetts, USA). ESR data were processed in MATLAB. All plots were generated using Origin 2019 (OriginLab Corporation, Northampton, Massachusetts, USA).

**Statistical analysis.** All numerical results are reported in the main text as average of repeated measurements, and the standard deviation represents the error. All measurements were repeated at least three times.

## Data availability
The authors declare that all data supporting the findings of this study are available within the paper and its Supplementary Information files. Raw data are available upon request from the corresponding author.

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

## Acknowledgements

We thank Prof. Jean-Noël Hyacinthe for useful discussion and Dr. Sebastian Meier for setting up "remote dissolution" experiments at the Chemistry Department. This work was supported by the Danish National Research Foundation (DNRF124); the European Union's Horizon 2020 research and innovation program under the Marie Sklodowska-Curie grant agreement no. 713683 (COFUNDfellowsDTU); and the Swiss National Fund under the SPARK grant agreement no. CRSK-2_190547 and Ambizione grant agreement no. PZ00P2_193276.

## Author contributions

A.C., J.K., M.H.L., and J.H.A.-L.: study conception and design. A.C., J.K., M.K., A.C.P., S.P., and M.H.L.: acquisition of data. A.C., M.K., and M.H.L.: analysis and interpretation of data. O.O., S.P., and F.V.: synthesis of non-persistent radical precursor. A.C. and M.H.L.: drafting of manuscript. A.C.P., J.H.A.-L., and M.K.: first revision. A.C. and M.H.L.: final revision.

## Competing interests

Professor Ardenkjær-Larsen is CEO of the startup company Polarize. Polarize sells dDNP equipment for pre-clinical studies. All other authors declare no competing interests.
