## [Peer Review File · Communications Chemistry]

Reviewers' comments:

Reviewer #1 (Remarks to the Author):

In their paper, Capozzi et al. demonstrate an approach of quenching UV-induced radicals in a DNP process for producing hyperpolarized [U-13C, d7]-D-glucose, storing it in a solid radical-free hyperpolarized state, thus, allowing transport of the hyperpolarized sample from the point of production to a remote point of NMR detection. While the idea of generating and quenching UV-based radicals in the dDNP process (as well as the idea of transportable DNP-polarized agents) is not new, realization of the separation between the two steps of the process, i.e., hyperpolarization/quenching and detection, is demonstrated for the first time to the best of my knowledge. This was accomplished by solving a magnitude of “technicalities” which is, nonetheless, constitute a typical bottleneck in experimental science. In particular, it was discovered that implementing permanent magnets inside the DNP probe was necessary to “shelter” hyperpolarization during the sample extraction; otherwise, relaxation is too fast at low fields. The paper is very well written and polished. While some minor jargon is present throughout the paper, it will still be interesting for the hyperpolarized NMR community as it brings significant insights to this specialized area of research. I recommend publication in Communication Chemistry.

Reviewer #2 (Remarks to the Author):

In this work the authors nicely present a method to preserve the hyperpolarized state of metabolic contrast agents outside the polarization instrument. This is a major achievement that may lift a major barrier in metabolic hyperpolarized magnetic resonance research and allow other players to enter the field. The authors nicely discuss the achievements and the problems still to solve. The technology is demonstrated on [U-13C, d7]-D-glucose, an agent with a relatively short lifetime compared to the main dDNP agent [1-13C]pyruvate and therefore more challenging.

I have reviewed the parts within my expertise. I am not an expert on magnetic field simulations or hardware design, including the design of NMR probes or permanent magnets. I would trust the authors on those due to their track record in designing and implementing such instrumentation and the level of detail given.

I only have minor comments as regards to the text.

Introduction

1. “characterized by high glucose uptake” is not clear in the context of this statement.
2. As regards to FDG-PET, please indicate more clearly the ionizing radiation to which patients are exposed to and the limitations on repeated examinations and use in certain patient populations.
3. “good spectral separation” – not clear
4. The authors place major focus on the comparability to FDG-PET operational considerations. In this regard:
 - a. FDG-PET uses a 2-deoxyglucose derivative, please indicate that the parallel agent to this in hyperpolarized MR would be a stable isotope labeled 2-deoxyglucose agent. Please cite DOI: 10.1038/s41598-019-56063-0 in this regard.
 - b. The first use of hyperpolarized [U-13C, d7]-D-glucose for MR imaging that parallels the FDG-PET examination (without metabolic pathway resolution) was reported several years ago and should be

cited, DOI: 10.1002/cmml.1497.

Results

5. The section "A "make it all" device" is not results. I would move this to the Methods in a separate section on the Description of the system. The same goes for the 1st paragraph of the section "Hyperpolarized sample with extended lifetime". The same goes for Figure 1 and Figure 6.

6. Movie S1 is important and well presented, maybe I missed it but how long did the transfer procedure take here actually? From the rest of the text (Discussion) it is not clear if the time to reach the dissolution site was 3 min or the dissolution procedure took 3 min from arrival to the dissolution site. I thought the latter as per the description in the Results but the paragraph starting with "A more potent source..." in the discussion confused me.

Discussion

7. What would be the role of the [U-13C, d7]-D-glucose formulation? The authors understandably use a formulation already used by this group. However, it should be noted that other formulations have been developed and studied for this agent (even if in a different magnetic field). For example, please see DOI: 10.1002/cphc.201900946.

8. Sentence starting with "Under these conditions, the T1 measured..." unclear.

Online methods

9. Dissolution: It is not clear why one would dissolve a glucose sample in a phosphate buffer as glucose is not acidic. The dissolution buffer appears hypo-osmotic and contains EDTA, both are likely to lead to prolonged T1 compared to solutions intended for biological use.

10. Enhancement calculation: 100 ms repetition time seems really short for 13C of glucose. Was this time enough for obtaining fully relaxed spectra? If not, is the T1 under these conditions known? Was the line-width affected by Gd doping? Could it be that this affected the polarization % that was determined?

11. Page 22: relaxion, correct

Reviewer #3 (Remarks to the Author):

The authors report a very important improvement to dDNP: the transfer of frozen samples with long T1. The present some modification to a DNP system that allows them to keep the sample at an elevated magnetic field to reduce relaxation losses. the sample is transferred to an NMR and detected. A few % polarization were observed on glucose.

This report is an essential progress that must be published.

Unfortunately, I have some issues with the scholarly presentation of the work. I find many

superfluous sentences, colloquialisms, unclear structures on the one hand, and little substantial data on the other (eg. on chemistry, its a chemistry journal after all). The abstract (which is not an abstract in my opinion) is even a bit misleading in suggesting that you solved the T1 issue of glucose. The short T1 in vivo remains the major issue which is not addressed at all (see paper by Rodrigues et al). You dont explicitly say that you did, but you don't deny either, and in the context it appears as such.

You will find many comments in the attached file, unclear language is highlighted.

Thus I strongly encourage the authors to revise this utterly important paper make to make it more matter-of-fact-style, to tone down many expressions and to give realistic assesement of Glucose. To be honest, I don't see Glucose going anywhere until T1 in vivo is longer, so it may not be the perfect molecule to demonstrate delivery of HP samples, but as a demonstrator its OK.

To be absolutely clear: this is an absolute breakthrough for DNP and must be published. But IMHO, please modify the way you present it.

Thanks for your efforts! its an important contribution to the field.

**Metabolic contrast agents produced from transported solid ¹³C-**
**glucose hyperpolarized via Dynamic Nuclear Polarization**

Andrea Capozzi,^{1,2*} Jan Kilund,² Magnus Karlsson,² Saket Patel,² Arthur Cesar Pinon,² Olivier
Ouari,³ Mathilde H. Lerche,² and Jan Henrik Ardenkjær-Larsen.²

¹ *LIFMET, Department of Physics, EPFL, Station 6 (Batiment CH), 1015 Lausanne*
*(Switzerland)..*

² *HYPERMAG, Department of Health Technology, Technical University of Denmark, Building*
*349, 2800 Kgs Lyngby (Denmark).*

³*Institut de Chimie Radicalire Aix-Marseille Université, CNRS, ICR UMR 7273, 13397 Marseille*
*Cedex 20 (France).*

**Corresponding author**

*Dr. Andrea Capozzi

EPFL SB IPHYS LIFMET

CH F0 632 (Bâtiment CH)

Station 6

CH-1015 Lausanne

17 T: +41 21 693 05 88

Email: andrea.capozzi@epfl.ch

ORCID: 0000-0002-2306-9049

**Abstract**

Hyperpolarized (HP) ^{13}C -labelled metabolic contrast agents (MCAs) via dissolution Dynamic
Nuclear Polarization (dDNP) are generating significant interest for their ability to inform, non-
invasively and in real-time, on tissue specific aberrant metabolism. However, an inherent short
lifetime of these agents combined with demanding and expensive hyperpolarization equipment
hamper the adoption of the method in the clinic. For these reasons, glucose metabolism for cancer
diagnostic and treatment monitoring purposes is currently performed by means of ^{18}F -fluoro-
deoxy-glucose (^{18}F -FDG) Positron Emission Tomography (PET) examinations. Nevertheless, this
technique presents some limitations such as lack of specificity in organs with a high normal
glucose uptake and use of ionizing radiation.

In this work, we present a paradigm shift in the dDNP technique built on photo-induced
thermally labile radicals, which allow solid sample extraction from the dDNP polarizer and hours
long lifetime of the MCAs. We demonstrate the ability to disconnect elaborate equipment to
produce above 10,000-fold signal enhanced MCAs, $[\text{U-}^{13}\text{C}, \text{d}_7]\text{-D-glucose}$, from its end-user site,
enabled by HP sample storage and transport. Such remote production of ^{13}C -labelled MCAs, with
37 hours long lifetime at appropriate transport conditions, would be much like the way ^{18}F -FDG
PET is currently performed in the clinic.

Introduction

Changes in metabolic pathways, causal of the origin or progression of diseases such as cancer,
diabetes and neurodegenerative diseases can be studied non-invasively with metabolic imaging
methods.¹⁻³ The latter are thus powerful means to diagnose and monitor response to therapy.⁴
Despite significant research progress, methods used to measure metabolism in patients are still
limited and blind to many cellular processes.⁵

Among those, ¹⁸F-fluoro-deoxy-glucose positron emission tomography (¹⁸F-FDG PET) is the
current benchmark to assess hypo- or hypermetabolism in clinical practice, in particular for what
concerns cancer diagnosis, staging, and treatment monitoring.⁶ Nevertheless, ¹⁸F-FDG PET has
its limitations. For instance, it suffers from poor specificity in organs with a high normal glucose
uptake;⁷ non-cancerous inflammations, characterized by high glucose uptake, can result in false
positives;⁸ the radioactive nature of the tracer exposes the patient to potentially dangerous
ionizing radiations. Most of all, the information obtained is limited to glucose uptake since
downstream metabolites like lactate and tricarboxylic acid (TCA) cycle intermediates are
invisible to the technique.^{9,10}

A more direct and specific way to track metabolism in vivo is to follow the fate of exogenous
substrates by ¹³C magnetic resonance spectroscopy (MRS) or spectroscopic imaging (MRSI).
These techniques allow phenotypical characterization of tumors by looking at downstream
metabolism enabled by good spectral separation of the different metabolites.¹⁰ However, ¹³C-
MRS and MRSI widespread use in the clinic is limited by low sensitivity. The MR signal is
proportional to the nuclear spin concentration and nuclear spin alignment (i.e. polarization) with
respect to the applied magnetic field. Both features are limited in this kind of experiments. Signal
averaging is the typical workaround to overcome low sensitivity. Unfortunately, the price to pay
is poor temporal resolution and thus access to the highly informative metabolic flux.¹¹

Deuterium metabolic imaging, a novel, noninvasive method, combines deuterium magnetic
resonance spectroscopic imaging with oral intake or intravenous infusion of 6,6-²H-labeled
glucose to generate three-dimensional metabolic maps.¹² Although straightforward to implement,
this technique is challenged by poor metabolite separation at clinical magnetic field strengths.

General advancement of MRS techniques comes from recent developments in hyperpolarization
technologies.¹³ Using hyperpolarized agents, the low sensitivity drawback of ¹³C-MRS can be
largely circumvented,¹⁴ and metabolic activity can be imaged non-invasively and in real-time in
humans.^{15,16}

Dissolution dynamic nuclear polarization (dDNP) is the most versatile among the
hyperpolarization methods.¹⁷ With dDNP, hyperpolarized (HP) metabolic contrast agents
(MCAs) can be obtained with ¹³C-nuclear spin polarization up to 100,000-fold compared to
thermal equilibrium on a clinical scanner. The MCAs are produced in an expensive and
technically demanding device called dDNP polarizer. The polarizer provides the appropriate
conditions of temperature and magnetic field to transfer polarization from unpaired electron
spins, added to the sample in form of organic radicals, to ¹³C nuclear spins in the MCA, by using
microwave irradiation. Unfortunately, dDNP is characterized by a striking imbalance between
sample throughput and lifetime of the HP state. It typically takes hours to create a single
injectable dose of MCA, whereas, after dissolution and extraction from the polarizer, the HP
MCA's lifetime is only minutes in the best case. Currently, to equip an MR facility with
hyperpolarization, the dDNP device must be located in near proximity to the scanner. Although
dDNP HP ¹³C-MRS has the potential to revolutionize diagnostic radiology enabling precision
medicine and personalized healthcare,^{15,19-21} this limitation threatens its development into
widespread clinical use.

The ¹³C polarization's half-life within the MCAs is several orders of magnitude longer when kept
frozen at cryogenic temperature. This allows, in principle, transportation of the MCAs far away

from their production site.²² Unfortunately, a dDNP sample cannot be extracted as a frozen solid
without losing its hyperpolarization.^{17,23} The problem is the paramagnetism of the radicals that
are added to the sample to allow the DNP process to take place inside the polarizer.²⁴ Indeed, in
solid samples doped with radicals, the nuclear spins relaxation becomes prohibitively fast at low
magnetic field.²⁵ These are the conditions experienced by an HP sample when lifted far away
from the high field of the DNP machine.^{22,26,27} This is the reason why, to keep the MCA's
hyperpolarization alive during sample extraction, the original dDNP technique requires the
dissolution to be performed inside the polarizer at high field.¹⁷

To dispense of the presence of technically demanding and costly hardware at individual clinical
sites, and instead hyperpolarize MCAs at a central facility for subsequent storage and distribution
would demand a radically new way of producing HP MCAs. Such remote production of ¹³C-
labelled MCAs would be much like the way clinical examinations are performed with ¹⁸F-FDG
PET, where the tracer is delivered on demand.

The two key challenges to address are to extract the MCA as a frozen solid from the polarizer,
while preserving its hyperpolarization, and to prolong the ¹³C HP signal lifetime as much as
possible under sample transport conditions. Three different concepts have been published for
producing long-lasting HP samples for storage and transportation.²⁶⁻²⁸ The critical point in these
concepts is the absence or drastic reduction of paramagnetic relaxation during extraction from the
polarizer of the sample still in the solid-state.

The first approach, proposed by Hirsch et al., does not use DNP. Thus, no paramagnetic agents
are added to the MCA formulation, which is hyperpolarized by brute force (e.g. cooling down the
sample to very low temperatures while keeping it at high magnetic field).²⁶ Although an easy
solution to get rid of detrimental paramagnetism, this method is very slow (i.e. it takes tens of
117 hours to thermally polarize the sample when working at 2 K and 14 T) and the obtained liquid-
118 state hyperpolarization is between two and three orders of magnitude lower with respect to DNP.

The second approach, introduced by Ji et al., physically separates the radicals from the ^{13}C -MCA
in the sample. DNP is performed on ^1H nuclei of a radical doped solvent that impregnates without
dissolving the MCA. Then, ^1H - ^1H spin diffusion and cross-polarization transfer the high spin
order from solvent protons to ^{13}C nuclei borne on the MCA molecule.²⁷ This is an elegant idea
for combining hyperpolarization via DNP to HP sample extraction and transportation. However,
the distance between the ^{13}C nuclei and the radicals in the sample matrix makes the DNP process
less efficient by a factor of three, at least.^{27,29} Moreover, the radical-rich phase of the sample
involves non-biocompatible solvents such as toluene and tetrahydrofuran.

The third approach hyperpolarizes the sample employing labile radicals originating from alpha
keto acids UV-light irradiation. These radicals are stable at low temperature, where the DNP
process takes place, but recombine into diamagnetic species above 190 K (i.e. the radical
quenching temperature).³⁰⁻³² Therefore, a radical free HP solid sample can be obtained by heating
the latter above the UV-radicals quenching temperature. Hyperpolarization of ^{13}C -MCAs using
photo-induced non-persistent radicals has been optimized and shown to perform as good as DNP
using stable radicals,^{30,32,33} while employing endogenous/biocompatible substances only.

In 2017, Capozzi et al.²⁸ demonstrated in a proof-of-principle study that labile radicals for dDNP,
generated via UV-irradiation of a sample containing a fraction of $[1-^{13}\text{C}]$ pyruvic acid, could be
quenched inside the polarizer by means of a “sample thermalization procedure”, while retaining
most of the hyperpolarization obtained via DNP in the solid state. A dramatic increase of the ^{13}C
T_1 was recorded after the removal of the radicals and this opened perspectives for solid sample
storage. Unfortunately, lack of controlled sample extraction allowed no real attempt of transport.

In this work we exploit UV-induced radicals to generate highly polarized ^{13}C -MCAs in the liquid
state with no need for dDNP polarizer on site. For the first time, we demonstrate transportation at
cryogenic temperature of such samples. We establish a robust protocol for sample loading,
polarization, thermalization, extraction, transport and dissolution away from the production site

of the HP ^{13}C -MCA solid sample. To this end we chose deuterated trimethylpyruvic acid (d_9 -
TriPA) as UV-radical precursor and $[\text{U-}^{13}\text{C}, \text{d}_7]$ -D-glucose as substrate (30), the latter being a
molecule showing increasing interest in the hyperpolarization community thanks to the richer
metabolic pathways it can give access to, compared to routinely used $[\text{1-}^{13}\text{C}]$ pyruvate.^{34–36}

Specific hardware development allowed us to solve two stringent physics problems: 1) efficient
heating of the sample while retaining most of the polarization; 2) avoid fast spin lattice relaxation
at low-field present even in absence of radicals. We implemented a custom designed fluid path
(CFP) with the purpose of diagnosing and solving experimental challenges as well as making it
possible to run all steps of the experiment in a user-friendly closed system.

**Results**

**A “make all” device**

Our first aim in this study was to build a device that could allow us to investigate, in a robust and
reproducible way, all steps involved in a “remote DNP” experiment employing UV-induced
radicals: UV-irradiated sample loading into the dDNP polarizer while keeping it below the
critical temperature of around 190 K, hyperpolarization of the sample, UV-radicals elimination,
HP sample extraction from the polarizer, HP sample storage and transport and finally HP sample
dissolution away from the production site. To this end, we developed what we call a “custom
designed fluid-path” (CFP). The device is reported in Fig. 1. All technical details for the
construction and operation of the CFP are described in *Methods*. The threaded vial (parts 5 and 6
in Figure 1A), sealed to superfluid He by compressing a PTFE O-ring (part 12 in Figure 1D)
allows loading solid samples through a 7 mm diameter opening. Moreover, this approach makes
the CFP reusable as far as the O-ring is replaced after each experiment. The top part of the device
is equipped with a quick release connector (part 1 in Figure 1A). This component made it
possible to both quench the radicals and later dissolve the sample by injecting He gas or hot
solvent inside the CFP, respectively. Moreover, the quick release helped transportability of the
CFP. Indeed, the CFP could be easily moved from the sample loading/leak-test station (see ref.
[32,37] and Figure S1 for details) to the polarizer and finally into the storage/transport unit. The
dynamic sealing (see Figure 1C) allowed us to operate the polarizer at low pressure (1 – 20 mbar
range) during all the experiment's steps. Moreover, it was an asset when investigating the
sample's relaxation properties, with and without radicals, at different distances from the
superconductive magnet's isocenter, while keeping the base temperature unchanged. Most
importantly, our aim was to deliver to the final user a compact “plug and play” solution to obtain
the MCA in the liquid state on site: a CFP inside an appropriate transportation device.

**Hyperpolarized sample with extended lifetime**

In order to generate a radical free HP sample, we designed a specific experimental procedure
involving the hardware earlier described. To this end, we used one single sample preparation
consisting in 2.2 M of [U-¹³C,₇D]-D-glucose dissolved in 60 μL of glycerol:water 1:1 (v/v). The
d₉-TriPA was added in amount corresponding to 10% of the final volume to generate 40±4 mM
or radical after UV-light irradiation (see *Methods* for details about sample preparation). The
preparation and polarization of the sample was developed and optimized in a former
publication.³⁰ Figure 2 shows the different steps and the NMR signal time course of a typical
hyperpolarization experiment followed by radical quenching. DNP was performed at 6.7 T and
1.20±0.05 K using a dDNP polarizer (Magnet and cryostat from Magnex Scientific Ltd, Yarnton,
UK) conceptually similar to the idea introduced in 2003,¹⁷ but equipped with a sample loading
chamber/air-lock module and a gate valve to be compatible with the fluid path technology.^{38,39}

**Figure 2A and the green portion of Figure 2D report the first part of the experiment:** the sample
vial was lowered into the NMR coil, microwave irradiation was performed at optimal conditions
(see *Methods* for details), the sample reached a solid-state ¹³C polarization of 45±5 %
approximately 1 h (buildup time constant 1300 ±10 s), in good agreement with our former
study.³⁰ Then, the radical quenching procedure started.

Figure 2B and the yellow portion of Figure 2D illustrate this step: microwaves were switched off,
the vial was lifted 15 cm above the NMR coil, outside the liquid He bath, and left there for 5 min;
the CFP quick release was connected to a He gas line and the gas blown towards the sample.

Figure 2C and the orange portion of Figure 2D describe the last step of the experiment: the vial
was moved back to the measurement position inside the NMR coil for checking the outcome of
the radical quenching procedure. Firstly, NMR was acquired to evaluate the polarization loss
during sample heating. Secondly, microwaves were switched ON again to verify the absence of
any DNP process and qualitatively verify the quenching of the radicals. A quantitative check was
performed later by extracting the CFP from the polarizer and recovering the beads from the
sample vial into liquid nitrogen to measure any residual radical concentration by ESR (see
*Methods*).

Radical scavenging parameters were optimized in a series of different experiments. The best
result was found by blowing room temperature He gas for 20 s at 6 bar of pressure. This
procedure allowed us to get rid of 99% of the radical in the sample (see Figure S2). At optimal
condition the polarization loss during the sample heating by means of He gas blowing was
around 20% of the initial value. The inset in Figure 2D shows the “signature” of a successful
thermalization experiment: the signal increased in the first few recorded NMR spectra.

Quenching of the radicals from the HP sample caused a dramatic increase of ^{13}C nuclear spin-
lattice relaxation time. Figure 3 reports the signal evolution as a function of time, at 4.2 K and 6.7
213 T, in absence of microwave irradiation for a quenched sample (black circles) and a sample with
214 the UV-radicals still present (blue circles). The ^{13}C T_1 increased from $2,300 \pm 20$ s to
215 $200,000 \pm 3,600$ s (i.e. 55 ± 1 h), confirming that the UV-radicals in the sample represented the
216 main source of relaxation. We performed these measurements at 4.2 K instead of 1.2 K to better
and more quickly visualize the T_1 difference between the two samples.

In a separate series of experiments, by implementing a manual field cycling inside the polarizer,
we also measured the ^{13}C relaxation of a sample after UV-radicals quenching at 4.2 K and 1 T
(see *Methods* for details about the field cycling implementation). In Figure S3 we report the
results: by fitting the data to a mono-exponential curve, we found a T_1 of 4.0 ± 0.5 h ($R^2 = 0.97$).

**Minimizing polarization loss during solid sample extraction**

Unfortunately, the UV-radical quenching procedure alone was not enough to extract the sample
from the DNP machine while retaining most of the polarization. In Figure 4A we outline the
NMR signal intensity of two HP samples (one with UV-radicals still active and one after
quenching) as a function of the polarizer's decreasing magnetic field along the z-axis. Thanks to
the flexibility of the CFP dynamic sealing, the sample extraction path was recorded in steps of 10
228 cm from the magnet's isocenter to the loading chamber. For each step, the sample vial was lifted
to the desired height, allowed to relax for 5 s and then lowered back to the measurement position
inside the NMR coil. The first HP sample was not subjected to quenching before extraction (blue
circles). In this case, a magnetic field of 350 mT (approx. 30 cm from the magnet's isocenter)
was enough to cause a loss of almost half of the polarization created via DNP. Exposing the same
sample to 100 mT (approx. 40 cm from magnet isocenter) caused an almost complete loss of
polarization. The second HP sample was subjected to quenching before investigating the
extraction. This sample could be exposed to a magnetic field as low as 40 mT (approx. 50 cm
from magnet isocenter) while retaining most of the polarization. However, lower values of the
magnetic field relaxed the polarization completely during the 5 s waiting time. The magnetic field
profile of the superconductive magnet was measured with a Hall probe up to 3 T and simulated in
MATLAB, according to the coil geometry, from 0 T to 6.7 T (see Figure S4). The simulation was
in good agreement with the measured data points ($R^2 = 0.98$).

These results gave us useful information about how to modify the original DNP probe⁴⁰ to shelter
the hyperpolarization of UV-radicals quenched samples during extraction. Accordingly, a

“magnetic rail” was designed using NdFeB permanent magnets along the path traveled by the
sample. The starting point of the inserted magnets was chosen to be just below the position where
the UV-radical quenched sample experienced its initial polarization loss (i.e. 40 cm above the
magnet isocenter). The arrangements of the permanent magnets generated a field perpendicular to
the one of the polarizer, and the value of the total field (from polarizer and permanent magnets)
never dropped below 100 mT. To achieve this, we used four Halbach array arrangements,
designed according to the probe geometry, to cover the space from 40 cm above the polarizer’s
isocenter to the loading chamber.

The additional field from the permanent magnets allowed us to move a thermalized sample from
inside the NMR coil to the loading chamber, while retaining more than 90% of the polarization
(see Figure 4B). The total magnetic field as a function of the distance from the magnet’s isocenter
is reported in Figure 4C. Details about the magnetic rail construction and magnetic field
simulation are reported in the *Methods* section and Figure S5 and S6, respectively. Placing
permanent magnets inside the DNP probe had no detrimental effects neither on the homogeneity
or shift of the NMR resonance nor on the polarizer base temperature, despite potentially
increased heat conductivity.

**Successful sample transport and straightforward remote dissolution**

Once we made sure we could freely move thermalized samples along the polarizer z-axis while
retaining most of the polarization, we investigated and implemented transport and remote
dissolution. From field cycling experiments inside the polarizer, it was clear that hours long T_1
could be obtained for [U- ^{13}C , d $_7$]-D-glucose at 1 T and liquid helium temperatures (see above).
Since storage in liquid helium requires a cryostat, we aimed instead at keeping the sample at
liquid nitrogen temperature in a field of 1 T. The transport and remote dissolution procedure is
outlined in Figure 5. The transport device was composed of two parts (see Figure 5A, B, C and
D): a 300 mT four elements Halbach array magnetic guide at the top and a 1 T eight elements

Halbach array storage magnet at the bottom (see Figure S6 for details). The transport device was
precooled to 77 K by placing it into a Styrofoam box filled with liquid nitrogen (see Figure 5E).
The HP sample transport and remote dissolution entailed four main steps. Once the HP
thermalized sample reached the loading chamber/air-lock, the polarizer's gate valve was closed,
and the loading chamber/air-lock disconnected from the rest of the DNP probe. The loading
chamber/air-lock was then docked to the transport device (Figure 5A and B), and the sample was
pushed down into liquid nitrogen to reach the storage magnet (Figure 5C and D). The Styrofoam
box was then put on a trolley, brought to a liquid state NMR laboratory installed two floors above
the dDNP polarizer location, and the CFP connected to a compact dissolution station (see Figure
S7) to extract the MCA in the liquid state. The HP MCA was finally injected into a 5 mm NMR
tube and inserted in a 9.4 T vertical magnet to measure the polarization (see ref. 37 and *Methods*
for details about the dissolution procedure).

The elapsed time between disconnection of the loading chamber/air-lock and remote dissolution
was approx. 3 min. The glucose liquid-state polarization after dissolution was 4.0 ± 1.0 %, ($n = 4$).
One last optimization of the experiment was done by replacing the loading chamber/air-lock
vacuum clamp with a quick release one, in order to speed-up its disconnection. This improved the
measured ^{13}C liquid-state polarization to 9% ($n = 1$) (results reported in Figure 5F). We
encourage the reader to watch the video recorded about the hyperpolarization transport and
remote dissolution (see Movie S1).

**Discussion**

In this study, we exploited UV-induced labile radicals for DNP and smart hardware design to
demonstrate, for the first-time, transport at cryogenic conditions and remote dissolution of HP
$[\text{U-}^{13}\text{C}, \text{d}_7]\text{-D-glucose}$. The need to thoroughly understand nuclear relaxation phenomena as a
function of temperature and field pushed us to develop a device, the CFP, able to cover all
different steps of the experiment in a controlled way. The CFP turned out to be an extremely

useful device also for direct dissolution DNP experiments.^{41,42} Implementation of permanent
magnets inside the DNP probe was a crucial step needed to successfully shelter the
hyperpolarization during sample extraction. In vision of a distribution of HP MCAs on a larger
scale, we envisage that several CFPs could be prepared by trained personnel and delivered on
demand.

At this stage of the study a careful evaluation of the polarization losses is important to suggest
further improvement of this technique as a game changer for production of transportable DNP
hyperpolarized MCAs.

As earlier reported,³⁰ a ^{13}C liquid-state glucose polarization of approx. 30% is obtained when the
sample formulation used in this study is dissolved directly from the dDNP polarizer (10 s interval
between dissolution onset and start of the NMR acquisition on the 9.4 T magnet, with a reported
$[\text{U-}^{13}\text{C}, \text{d}_7]\text{-D-glucose}$ T_1 in solution of 20 s). When compared to our best result so far (^{13}C
polarization of 9% on a remote dissolved sample 3 min after sample extraction), we lose 2/3 of
the polarization during the extraction and transport process.

We characterized one source of relaxation in the experiment. The UV-radicals quenching process
accounts for a relative polarization loss of 20%. This would project the maximum achievable
liquid-state ^{13}C polarization for glucose to 24%, if dissolution occurred right after this step of the
experiment. According to the data reported in Figure 4B, lifting a UV-radical quenched sample to
the loading chamber causes almost no loss of polarization, as confirmed by the study from Ji et
al. for glucose embedded in a porous matrix and not in direct contact with the radicals.²⁷
Moreover, if the gate valve was opened and the sample subjected to the cold He gas stream,
performing a fast extraction (10 s) compared to a slow one (approx. 2 min) did not make any
difference.

A more potent source of polarization loss is the unavoidable relaxation of HP glucose when kept
at 77 K and 1 T for the 3 min necessary to reach the site where the dissolution took place.

Unfortunately, we did not find in the literature T_1 data for glucose at such experimental
conditions and, for the time being we do not dispose of the needed hardware to perform these
measurements. We can, however, provide a rough estimation of the T_1 at these transport
conditions (1 T, 77 K) by referring to data published by Hirsch et al. for solid [1- ^{13}C]pyruvic
acid, at 1.3 T and 60 K, with no radicals present in the sample.⁴³ Under these conditions, the T_1
measured over 5 min for a partially annealed pyruvic acid sample and remained constant down to
20 K,⁴³ where the methyl groups rotation is supposed to be minimal.⁴⁴ Assuming a similar value
of T_1 for our sample at 1 T and 77 K, this would account for a relative polarization loss of
approx. 50% during transport, projecting the maximum achievable liquid-state ^{13}C polarization
value for glucose to 12 - 13%.

[revised manuscript text omitted]

probe: top part front view in Figure 6A, top part top view in Figure 6B, bottom part front view in
Figure 6C, bottom part section view in Figure 6D, top part section view in Figure 6E. Refer to
this figure for the following description.

Differently from Ji et al.,²⁷ we decided to equip the probe with NdFeB permanent magnets
(Supermagnete, Gottmadingen, Germany) along the path travelled by the sample instead of
winding the sample vial inside a small solenoidal electro-magnet to simplify extraction
operations.

To cover with permanent magnets the full path's length experienced by the sample during
extraction, we designed four different Halbach array arrangements. A four elements Halbach
array (7.5 mm x 7.5 mm x 100 mm bar magnets), held in place by stainless steel squared profiles
(Figure 6A, components 8), surrounds the probe stem (Figure 6C, components 12). The last four
elements of this "magnetic rail" enter the top flange by 90% of its thickness (see Figure 6E,
bottom inset). To fill the magnetic field gap between the ISO-KF-100 flange (Figure 6A,
component 5) and the mini gate-valve (Figure 6A, component 3), four octagonal Halbach arrays
(1.5 mm x 1.0 mm x 5 mm bar magnets) are stacked inside the two KF-16 half nipples (see
Figure 6E, component 18). The mini gate-valve volume is covered by gluing on it a two
elements Halbach array (30 mm x 30 mm x 10 mm block magnets) (see Figure 6E, right inset). A
second stack of three octagonal Halbach arrays covers the gap between the mini gat-valve and
loading chamber (Figure 6E, component 17). Finally, a 3D printed hexagonal Halbach array (12
460 mm x 12 mm x 12 mm block magnets) is placed around the bottom part of the loading chamber
(Fig. 6 A, component 2) to insure sufficiently high magnetic field during transfer of the sample
from the polarizer to the storage vessel. Details about the different Halbach arrays field profile
are reported in Figure S5.

**UV-sample preparation and handling**

All chemicals were purchased from Sigma-Aldrich (Brøndby, Denmark) excepted for the radical
precursor deuterated trimethylpyruvic acid (d₉-TriPA) that was synthesized in house. We worked
with a single kind of sample whose preparation was optimized in a former publication.³⁰ [U-¹³C,
d₇]-D-glucose was dissolved in 60±3 μL of glycerol:water 1:1 (v/v) to obtain a final glucose
concentration of 2.2M; d₉-TriPA was then added in amount corresponding to 10% of the final
volume; the solution was sonicated at 40°C for 5 min to efficiently degas the sample and improve
the glass quality after freezing. A solution volume of 6.0±0.3 μL was poured in liquid nitrogen as
a drop to form a frozen bead. The operation was repeated 10 times. The frozen sample was
transferred to a quartz Dewar (Magnetech, Berlin, Germany) filled with liquid nitrogen for UV
irradiation. The irradiation set up was extensively described earlier.³² UV-light was shined on the
sample for 300 s using a broad-band source (Dymax BlueWave 75, Connecticut, U.S.) at full
power (i.e. 19 W/cm²). Refer to Figure 7A for a simplified sketch of the setup. Radical
concentration was measured immediately after irradiation by inserting the tail of the quartz
Dewar in the cavity of an X-band spectrometer (Miniscope MS 5000, Magnetech, Berlin,
Germany) and following methods described earlier.³² Finally, the irradiated frozen beads were
loaded inside the CFP vial bottom part (Figure 7B, component 4). While keeping it and the
bottom wrench (Figure 7B, component 5) in a Styrofoam box filled with liquid nitrogen, frozen
pellets were transferred, a new PTFE O-ring (Figure 7B, component 3) put in place and squeezed
by screwing the vial top part (Figure 7B, component 2) by means of the top wrench (Figure 7B,
component 1). A leak test was then performed pressurizing the CFP with helium gas (see ref. 32
and former section for details).

Sample loading inside the polarizer proceeded as follows. The polarizer variable temperature
insert (VTI) was kept filled with 10 cm of liquid He and at low pressure, the CFP was
disconnected from the leak-test station (see Figure S1) and the vial (Figure 1A, components 5
and 6) quickly displaced from the Styrofoam box filled with liquid nitrogen to the loading
chamber, while flushing the latter with He gas. The dynamic sealing (Figure 1A, component 4)

[revised manuscript text omitted]

**Acknowledgments**

We thank Prof. Jean-Noël Hyacinthe for useful discussion and Dr Sebastian Meier for setting up
“remote dissolution” experiments at the Chemistry Department. This work was supported by the
Danish National Research Foundation (DNRF124); the European Union's Horizon 2020 research
and innovation programme under the Marie Skłodowska-Curie grant agreement no. 713683
(COFUNDfellowsDTU); and the Swiss National Fund under the SPARK grant agreement no.
CRSK-2_190547 and Ambizione grant agreement no. PZ00P2_193276.

**Author contributions**

Capozzi, Kilund, Lerche and Ardenkjær-Larsen study conception and design. Capozzi, Kilund,
Karlsson, Pinon, Patel, Lerche acquisition of data. Capozzi, Karlsson, and Lerche analysis and
interpretation of data. Ouari and Patel synthesis of non-persistent radical precursor. Capozzi and
Lerche drafting of manuscript. Ouari, Ardenkjær-Larsen and Karlsson critical revision.

**Competing financial interests**

Prof. Ardenkjær-Larsen is CEO of the startup company Polarize. Polarize sells dDNP equipment
for pre-clinical studies

**Additional information**

The supplementary material listed below is available for this paper online.

Figure S1. Leak-test station.

Figure S2. Radical leftover after quenching procedure.

Figure S3. Sample relaxation after thermalization at 4.2 K and 1 T.

Figure S4. Measurements and simulation of the polarizer stray magnetic field.

Figure S5. Finite elements simulations of the different Halbach arrays placed inside the new DNP
probe. Part1.

Figure S6. Finite elements simulations of the different Halbach arrays placed inside the new DNP
probe. Part2.

Figure S7. Finite element simulation for storage magnet.

Figure S8. Compact dissolution station head.

Movie S1. Hyperpolarization transport at storage conditions of 77 K and 1 T

Reprints and permissions information is available at www.nature.com/reprints. Correspondence
and requests for materials should be addressed to email: andrea.capozzi@epfl.ch. Raw data are
available upon request.

**References**

[revised manuscript text omitted]

Reviewer #2 (Remarks to the Author):

In this work the authors nicely present a method to preserve the hyperpolarized state of metabolic contrast agents outside the polarization instrument. This is a major achievement that may lift a major barrier in metabolic hyperpolarized magnetic resonance research and allow other players to enter the field. The authors nicely discuss the achievements and the problems still to solve.

The technology is demonstrated on [U-13C, d7]-D-glucose, an agent with a relatively short lifetime compared to the main dDNP agent [1-13C]pyruvate and therefore more challenging.

I have reviewed the parts within my expertise. I am not an expert on magnetic field simulations or hardware design, including the design of NMR probes or permanent magnets. I would trust the authors on those due to their track record in designing and implementing such instrumentation and the level of detail given.

I only have minor comments as regards to the text.

Introduction

1. “characterized by high glucose uptake” is not clear in the context of this statement.

We have modified the sentence which now reads: “and in inflamed normal tissue with the risk of false positives”.

2. As regards to FDG-PET, please indicate more clearly the ionizing radiation to which patients are exposed to and the limitations on repeated examinations and use in certain patient populations.

We have removed a sentence and two references to put less emphasis on PET-FDG which is in any case not a target of investigation in this paper. We modified another sentence. The relevant sentence now reads: “The use of ¹⁸F-FDG PET agents exposes the patient to gamma-rays, which restricts its use (e.g. in certain patient groups and for repeated examinations).”

3. “good spectral separation” – not clear

We have modified the sentence which now reads: “These techniques allow characterization of tumors by measuring downstream metabolism enabled by difficult signal disentanglement of the different metabolites”

4. The authors place major focus on the comparability to FDG-PET operational considerations. In this regard:

a. FDG-PET uses a 2-deoxyglucose derivative, please indicate that the parallel agent to this in hyperpolarized MR

would be a stable isotope labeled 2-deoxyglucose agent. Please cite DOI: 10.1038/s41598-019-56063-0 in this regard.

We agree with the reviewer that there has been put too much emphasis on the comparability to FDG-PET. We have modified both the abstract and also the introduction to reflect that this paper is not in fact a study that concerns such comparison. Accordingly, we will not include the suggested paper since it is no longer relevant.

b. The first use of hyperpolarized [U-13C, d7]-D-glucose for MR imaging that parallels the FDG-PET examination (without metabolic pathway resolution) was reported several years ago and should be cited, DOI: 10.1002/cmml.1497.

We have included this reference (now ref. 36) along with the other cited references concerning [U-13C, d7]-D-glucose.

Results

5. The section “A “make it all” device” is not results. I would move this to the Methods in a separate section on the Description of the system. The same goes for the 1st paragraph of the section “Hyperpolarized sample with extended lifetime”. The same goes for Figure 1 and Figure 6.

We agree with the reviewer that “a make it all device” sounds like a method section and it is somewhat redundant with “Custom designed fluid path” paragraph in the method. Nevertheless, this was a key development for this project. Therefore, we would like to keep Fig. 1 where among the results to give emphasis to this tool, but we rewrote the concerned paragraph keeping it short and limited to results only. We also changed the heading to “CFP performance” to be compliant to comment of reviewer 2. The heading “Hyperpolarized sample with extended lifetime” was also eliminated and the corresponding paragraph was included in “CFP performance”. Differently Figure 6 is already part of the methods. Because of the format of the journal (Methods after Results), in the “CFP performance” paragraph we kept only the fundamental “Methods information” to facilitate the reader understanding (e.g. sample composition and DNP working conditions). The new paragraph writes:

“CFP performance

The CFP (see Figure 1, all technical details are reported in *Methods*) allowed us to investigate, in a robust and reproducible way, all steps involved in a “remote DNP” experiment employing UV-induced radicals: UV-irradiated sample loading into the dDNP polarizer while keeping it below the critical temperature of around 190 K, hyperpolarization of the sample, UV-radicals elimination, HP sample extraction from the polarizer, HP sample storage and transport and finally HP sample dissolution away from the production site.

After 5 min of UV-light irradiation 40 ± 4 mM ($n = 3$) of radical was generated into the solid sample (see *Methods* for details about sample preparation). During sample loading into the polarizer the pressure increased from the base

pressure of 1 mbar to 10 mbar and went back to the initial value within 1 min. Performing DNP at 6.7 T/1.20±0.05 K under optimal microwave irradiation, ¹³C nuclei could reach a solid-state polarization of 45±5 % with a buildup time constant 1300 ±10 s (n = 3), in good agreement with our former study.³⁰ The polarization step of the experiment is sketched in Figure 2A, and a typical DNP buildup curve of the sample is reported in the green portion of Figure 2D.

The quenching step is sketched in Figure 2B. Before blowing He gas onto the sample for 20 s at 6 bar of pressure, best results were obtained by first switching OFF the microwaves, then lifting the vial 15 cm above the NMR coil, outside the liquid He bath, and leaving it there for 5 min (see absence of recorded NMR signal in the yellow portion of Figure 2D). The different parameters (i.e. blowing time, blowing pressure, quenching position and waiting time) of the procedure were optimized in repeated experiments. This allowed us to get rid of 99% of the radical in the sample (see Figure S2), while measuring a polarization loss of 20% of the initial polarization values (see orange portion of Figure 2D), when reinserting the vial into the NMR coil (see Figure 2C). The inset in Figure 2D shows the “signature” of a successful thermalization experiment: the signal increased in the first few recorded NMR spectra.

Quenching of most of the radicals was confirmed by the absence of any DNP process when switching the microwaves back ON, and it caused a dramatic increase of ¹³C nuclear spin-lattice relaxation time. The latter, measured at 4.2 K and 6.7 T, increased from 2,300 ±20 s for a non-quenched sample to 200,000±3,600 s (i.e. 55±1 h) for a quenched sample (see Figure 3), confirming that the UV-radicals represented the main source of relaxation.

In a separate series of experiments, by implementing a manual field cycling inside the polarizer, we also measured the ¹³C relaxation of a sample after UV-radicals quenching at 4.2 K and 1 T (see *Methods* for details about the field cycling implementation). In Figure S3 we report the results: by fitting a mono-exponential curve to data, we found a T_1 of 4.0±0.5 h ($R^2 = 0.97$).“

6. Movie S1 is important and well presented, maybe I missed it but how long did the transfer procedure take here actually? From the rest of the text (Discussion) it is not clear if the time to reach the dissolution site was 3 min or the dissolution procedure took 3 min from arrival to the dissolution site. I thought the latter as per the description in the Results but the paragraph starting with “A more potent source...” in the discussion confused me.

Yes, it was the time it took to go from the polarizer to the dissolution site. The dissolution procedure and transfer of the HP solution took 10 s as usual.

Discussion

7. What would be the role of the [U-13C, d7]-D-glucose formulation? The authors understandably use a formulation already used by this group. However, it should be noted that other formulations have been developed

and studied for this agent (even if in a different magnetic field). For example, please see DOI: 10.1002/cphc.201900946.

We chose this formulation for the high achievable polarization that glucose can reach when using UV-induced radicals. The work suggested by the reviewer, although of relevance at the more common field of 3.35 T, it concerns the trityl radical together with Gd doping. Trityl is a permanent radical and this kind of samples cannot be transported. Therefore, we will not include the suggested reference

8. Sentence starting with “Under these conditions, the T1 measured...” unclear.

We specified the concerned sentence by indicating clearly the value of T1 we estimate (i.e. 5 min at 77 K and 1 T).

Online methods

9. Dissolution: It is not clear why one would dissolve a glucose sample in a phosphate buffer as glucose is not acidic. The dissolution buffer appears hypo-osmotic and contains EDTA, both are likely to lead to prolonged T1 compared to solutions intended for biological use.

While we agree with the reviewer that the buffer used in the glucose demonstration is not going to be the choice in a clinical injectable, in particular since, as the reviewer points out, the injectable needs to be isotonic. In the present study, however, the demonstration did not hold such limitation and we chose to use a standard buffer for our ¹³C MCA studies where EDTA is added to prevent metal ions released in the heated boiler to impact the T₁ negatively. The phosphate buffer has no impact on glucose T1.

10. Enhancement calculation: 100 ms repetition time seems really short for ¹³C of glucose. Was this time enough for obtaining fully relaxed spectra? If not, is the T1 under these conditions known? Was the line-width affected by Gd doping? Could it be that this affected the polarization % that was determined?

We did not observe any substantial broadening from Gd doping on ¹³C; actually TR is 1.1 s (1s of FID acquisition + 0.1 s delay). Taking into account a measured T1 of glucose in presence of Gd (Omniscan) of 0.4 s and a flip angle of 5 deg, the error on the thermal equilibrium signal is below 1%. We updated in methods the TR to be 1.1 s.

11. Page 22: relaxation, correct

OK

Reviewer #3 (Remarks to the Author):

The authors report a very important improvement to dDNP: the transfer of frozen samples with long T1. They present some modification to a DNP system that allows them to keep the sample at an elevated magnetic field to reduce relaxation losses. The sample is transferred to an NMR and detected. A few % polarization were observed on glucose.

This report is an essential progress that must be published.

Unfortunately, I have some issues with the scholarly presentation of the work. I find many superfluous sentences, colloquialisms, unclear structures on the one hand, and little substantial data on the other (eg. on chemistry, it's a chemistry journal after all). The abstract (which is not an abstract in my opinion) is even a bit misleading in suggesting that you solved the T1 issue of glucose. The short T1 in vivo remains the major issue which is not addressed at all (see paper by Rodrigues et al). You don't explicitly say that you did, but you don't deny either, and in the context it appears as such.

We have modified the abstract to a concise presentation of motivation and the findings of the paper:

“Hyperpolarized (HP) ^{13}C -labelled metabolic contrast agents (MCAs) via dissolution Dynamic Nuclear Polarization (dDNP) can, non-invasively and in real-time, report on tissue specific aberrant metabolism. However, a short signal lifetime of these agents combined with the need to invest in demanding and expensive hyperpolarization equipment hamper the adoption of the method in the clinic.

In this work, we provide a robust methodology that allows remote production of the hyperpolarized ^{13}C -MCA. The methodology, built on photo-induced thermally labile radicals, allows solid sample extraction from the dDNP polarizer and hours long lifetime of the ^{13}C -MCAs at appropriate experimental conditions. We demonstrate the ability to disconnect the elaborate HP equipment from its end-user site. Exemplified with $[\text{U-}^{13}\text{C}, \text{d}_7]\text{-D-glucose}$, we remotely produce above 10,000-fold signal enhancement on the ^{13}C -MCA at 9.4 T, enabled by HP sample storage, transport and on-site dissolution.”

You will find many comments in the attached file, unclear language is highlighted.

Thus I strongly encourage the authors to revise this utterly important paper to make it more matter-of-fact-style, to tone down many expressions and to give a realistic assessment of Glucose. To be honest, I don't see Glucose going anywhere until T1 in vivo is longer, so it may not be the perfect molecule to demonstrate delivery of HP samples, but as a demonstrator it's OK.

We agree that glucose as well as most other hyperpolarized molecules are challenged by a short T_1 *in vivo* and that this paper only addresses the path from production to injection of the MCA. We have modified the text to make this point very clear.

We do however believe that the point made by the reviewer not only applies to glucose but is a general issue for the hyperpolarization technique. Rodrigues et al. reports an apparent *in vivo* T_1 of 9s. This value is similar to the apparent *in vivo* T_1 of pyruvate (approx. 12 s) at the same field strength and in mice. While pyruvate has a short T_1 already in blood due to unfavorable interactions, glucose is not in the same way affected in blood but is generally taken up and converted in all cell types leading to the short apparent T_1 .

On the other hand, outside the animal or human body the differences in T_1 between these two MCA's is large (approx. 14s (glucose) versus 60s (pyruvate) subject to specific field strength and temperature. To stand a chance as a clinical MCA it is thus needed to make the time between dissolution and injection as short as possible for glucose. The present paper addresses some of the challenges concerning this time frame. Today it takes, in the clinical setting, approx. 1 min between dissolution from the clinical HP equipment and injection of pyruvate into a patient. This time delay is the result of an evaluation of the injectable (pH, radical removal and transport from the equipment). Glucose is neutral and will not need a pH evaluation; the toxicology profile of the UV radical precursor is not addressed in this paper, but it is no longer a radical when it is injected; the dissolution from a small transport device will allow short proximity to the clinical MR scanner. All of the previous is likely to provide the possibility that glucose may in fact stand a chance to be injected as a highly polarized MCA.

For all these reasons we felt that glucose was the right choice as a demonstration molecule, however we agree with the reviewer that pyruvate may be a better choice when we in a next step will make a demonstration in a clinical setting.

Nevertheless, implementing all these comments into the introduction would be beyond the scope of the manuscript. Therefore, we included the following paragraph after mentioning the kind of sample used in this study:

“Moreover, the shorter liquid state T_1 of glucose compared to pyruvic acid after dissolution, would greatly benefit from quicker handling time prior to injection *in vivo* by reducing as much as possible the distance between the scanner and the dissolution device. This is far from trivial when dissolving the sample directly from the dDNP polarizer.”

Finally, we added the following sentence in the conclusions:

“Finally, we want to draw the attention of the reader to the fact that our new methodology can pave the way towards transportation of HP MCAs in the solid state, dispensing from the presence of a DNP polarizer at individual clinical sites. After transportation, the HP sample still needs to be dissolved. Because of the compact dimension of

the transport/storage device, this operation could be performed on the side of the MRI scanner reducing the handling time of the HP solution. Nevertheless, all dDNP limitations related to the MCAs' relaxation time in solution still stand"

To be absolutely clear: this is an absolute breakthrough for DNP and must be published. But IMHO, please modify the way you present it.

Thanks for your efforts! its an important contribution to the field.

Point-by-point for rev 3

1) Abstract is long and misleading

We rewrote the abstract (see above the response to general comment)

Let's write a more concise abstract

2)line 47, add a comma

OK

3) line 48, replace latter

"Latter" replaced with "These": "These techniques represent powerful means to diagnose and monitor response to therapy."⁴

4) line 52, glucose is not the golden standard for metabolism in general

We toned down the emphasis on the glucose. Please, see reviewer 2 answer 2.

5) line 59, rephrase glucose downstream metabolism issue in the context of PET

We followed the referee suggestion to remove "since".

6) line 62, what is a phenotypical characterization

We have chosen to delete phenotypical and rewrite the sentence. It now reads: "These techniques allow characterization of tumors by measuring downstream metabolism enabled by good spectral resolution of the different metabolites"

7) line 64, rephrase low sensitivity issue for NMR

"Sensitivity" was replaced with "SNR".

8) line 66, MR signal description not clear and sentence not clear.

We followed the reviewer's suggestion and rephrased as follows

"The MR signal is proportional to the nuclear spins' concentration and polarization (i.e. the net alignment of the nuclear spins ensemble in the direction of the applied magnetic field, the so-called B_0). Because of its gyromagnetic ratio, ^{13}C sensitivity is a fourth compared to proton MRS and its natural abundance is only 1 %."

9) line 67, averaging \rightarrow low temporal resolution not true

We are sorry, but we have to disagree with the reviewer here. Temporal resolution is linked to the repetition time of the NMR sequence: if we acquire 1 FID every 20 s our temporal resolution is 20 s. Indeed, in traditional (non hyperpolarized) ^{13}C MRS, where you have to average to get a decent signal, when investigating a metabolic pathway, you have information about what enters the pathway and what exits the pathway. Therefore, you try to model what happens in between. With hyperpolarized MRS you can "see" what happens in between because you can measure single shot (no average) spectra every e.g. 1s. We agree with the reviewer that averaging does not mean poor spatial resolution, but we did not mention that in the text.

10) line 73, general advancement \rightarrow correct sentence?

We rephrased the sentence that now reads: Limitations of MRS techniques has benefitted from developments in hyperpolarization technologies.

11) line 74, replace sensitivity with SNR

OK

12) line 75, remove largely

OK

13) line 78, specify "method to hyperpolarize small molecule in solution"

OK

14) line 80, specify field or polarization

We specified clinical scanner 1.5 T – 3 T.

15) line 81, rephrase, too colloquial

We replaced "called" with "known as".

16) line 84, add how the nuclear polarization is achieved

We completed the sentence with “Shining microwaves slightly at a frequency slightly higher or lower with respect to the electron spins resonance (ESR), polarization can be transferred from the electrons to the nuclei, thanks to their dipolar coupling.”

17) line 84, is there a connection between long polarization time and short life time

No, it is an observation and most of all a limitation of the technique. We have modified the sentence not to indicate any connection. It now reads: “Whereas, it typically takes hours to create a single injectable dose of MCA, the HP MCA’s lifetime is only minutes after dissolution and extraction from the polarizer”.

18) line 93, when cooling is the half-life longer also with radical present?

Actually, even in presence of radicals the T1 is much longer when lowering the temperature (look for instance at relaxation times for MAS DNP – 100 K- and dDNP – 1.2 K-). We clarified the sentence including the following: “ The ^{13}C polarization’s half-life within the MCAs is several orders of magnitude longer when kept frozen at cryogenic temperature, even in presence of radicals.”

19) line 101, condense above paragraph

We condensed the paragraph. It now reads: The ^{13}C polarization’s half-life within the MCAs is several orders of magnitude longer when kept frozen at cryogenic temperature, even in presence of radicals. This allows, in principle, transportation of the MCAs far away from their production site.²² Unfortunately, a dDNP sample cannot be extracted as a frozen solid without losing its hyperpolarization.^{17,23} The problem is the paramagnetism of the radicals that are added to the sample to allow the DNP process to take place inside the polarizer,²⁴ which induce nuclear spins relaxation that becomes prohibitively fast at low magnetic field.²⁵ These are the conditions experienced by an HP sample when lifted far away from the high field of the DNP machine.^{22,26,27}

20) line 104, rephrase...too dramatic

We undramatized and rephrased: “Lifting the mandatory presence of technically demanding and costly hardware at individual clinical sites could be realized, instead, if HP MCAs were produced at a central facility for subsequent storage and distribution to the site of action. Such remote production of ^{13}C -labelled MCAs could be envisioned to be much like the way clinical examinations are performed with ^{18}F -FDG PET, where the tracer with a short lifetime is delivered on demand.”

21) line 113, sentence sounds colloquial

We modified the sentence as follows: The first approach, proposed by Hirsch et al., does not use DNP to increase the polarization of the substrate of interest. Indeed, no paramagnetic agents are added to the MCA formulation, which is hyperpolarized by brute force (e.g. cooling down the sample to very low temperatures while keeping it at high magnetic field).²⁶

22) line 154, make it all is colloquial

We modified the heading as follows: “CFP performance” and shortened the paragraph to show results only (see reviewer 2, comment 5).

23) line 155 to 176, not results

See point 22 and referee 2

24) line 185, please describe what is on the figure

The concerned paragraph was modified and Figure description addressed directly. See point 22 and referee 2.

25) line 205 to 208, rephrase and be more precise with experimental conditions

We modified the paragraph to be more precise and provide only information useful to the understanding of the work (see point 22)

26) line 217, reasoning not clear rephrase

We added the following paragraph to methods (**Microwave delivery and solid-state NMR measurements**):
“Relaxation after thermalization was acquired by pulsing every 10 min. These measurements were performed at 4.2 K instead of 1.2 K because even in presence of radicals the relaxation time at 1.2 K can be several hours long, making it difficult to interpret the outcome of the quenching procedure. Differently, at 4.2 K amorphous solids enter a different relaxation regime (from direct process to Raman and Orbach, see Tom Wenckebach book on “Essential of Dynamic Nuclear Polarization”), and the T₁ becomes tens of minutes long when radicals are present. In absence of radicals the T₁ increases to several hours at 4.2 K, making it straightforward to interpret the outcome of the quenching procedure.”

27) line 221 rephrase the fitting sentence

Thank you for spotting the mistake, we modified the sentence accordingly:

“In Figure S3 we report the results: by fitting a mono-exponential curve to data, we found a T₁ of 4.0±0.5 h (R² = 0.97).”

28) line 222, change the heading, it is misleading

We changed it as follows: “**Radical free solid sample extraction**”

29) line 224, rephrase

We followed the reviewer’s suggestion and rewrote the paragraph as follows. All non-essential information was moved to methods:

“Radical free solid sample extraction

Despite quenching the radicals prior to HP solid sample extraction reduced the polarization losses from 90 % to 10 %, when exposing it to a magnetic field as small as 40 mT, lower values made the polarization to relax completely (see Figure 4A).

As the above results indicate severe relaxation due to exposure of the sample to a magnetic field lower than 40 mT, we modified the original DNP probe³⁹ by adding a “permanent magnets rail” providing a magnetic field of least 100 mT and oriented perpendicularly to the polarizer B_0 (see Figure 4B). Details about the magnetic rail construction and magnetic field simulation are reported in the *Methods* section and Figure S5 and S6, respectively.

Repeating the experiment employing the new DNP probe, we were able to move a quenched sample from the polarizer isocenter to the loading chamber while retaining more than 90% of the polarization (see Figure 4C).

It is important to notice that placing permanent magnets inside the DNP probe had no detrimental effects neither on the homogeneity or shift of the NMR resonance nor on the polarizer base temperature, despite potentially increased heat conductivity.”

30) line 225, make it more clear

See comment 29.

31) line 236, “most is not precise enough”, provide numbers

See comment 29.

32) line 240, please condense the paragraph above

See comment 29.

33) line 242, modify sentence to more informative and concise:

See comment 29.

34) line 239 and 243, move information to methods

OK

35) line 249, redundant

OK. We removed the sentence “to cover the space from 40 cm above the polarizer’s isocenter to the loading chamber.”

36) line 259, change heading to less colloquial jargon

We modified the heading as follows: **“Sample transport and remote dissolution”**

37) line 266, no results until here

We followed the reviewer suggestion and condensed the paragraph as follows. All other information was moved to methods:

“Sample transport and remote dissolution

From field cycling experiments inside the polarizer, it was clear that hours long T_1 could be obtained for [U- ^{13}C , d $_7$]-D-glucose at 1 T and liquid helium temperatures (see above). Since storage in liquid helium requires construction of a cryostat, we obtained the first results at liquid nitrogen temperature in a field of 1 T employing a simple transportation device (see Figure 5A to E and *Methods* for details about the construction of the transportation device).

Disconnecting the loading chamber containing the sample, lowering it into liquid nitrogen inside the storage magnet and reaching a NMR spectrometer placed 50 m far away from the polarizer took approx. 3 min. Once close to the NMR spectrometer, on-site dissolution generated a glucose polarization of $4.0 \pm 1.0\%$, ($n = 4$). One last optimization, aiming at speeding up the loading chamber disconnection, concerned the replacement of its vacuum clamp with a quick release one (results reported in Figure 5F). We encourage the reader to watch the video recorded about the hyperpolarization transport and remote dissolution (see Movie S1).”

38) line 271, gate valve is laboratory slang?

We chose to keep the term “gate valve” since this is the technical name on the market for this device.

39) line 277, still no results

See point 37

40) line 279, I suggest to condense the above paragraph and move to methods what it is not results

See point 37

41) line 288, remove “smart”

“smart” was replaced with “new”

42) line 291, remove your motivation

We removed the sentence.

43) line 318, can you condense the previous paragraph?

Although being concise is important, we think that an exhaustive discussion about the polarization losses is crucial. Nevertheless, we tried to polish the text to the best of our capability. Now the paragraph reads:

“We characterized one source of relaxation in the experiment. The UV-radicals quenching process accounts for a relative polarization loss of 20%. This would project the maximum achievable liquid-state ^{13}C polarization for glucose to 24%. According to the data reported in Figure 4B, lifting a UV-radical quenched sample to the loading chamber causes almost no loss of polarization.

Moreover, if the gate valve was opened and the sample subjected to the cold He gas stream, performing a fast extraction (10 s) compared to a slow one (approx. 2 min) did not make any difference.”

44) line 325, what value of T1

We specified the sentence by indicating clearly the value of T1 we estimate (i.e. 5 min at 77 K and 1 T)

45) line 328, comment on how to measure signal loss due to heating during docking

Although, running a series of experiments, it could be possible to estimate this loss e.g. by leaving the sample into the loading chamber, for increasing time intervals followed by dissolution and measurement in the liquid state, this would not represent a sufficiently controlled experimental environment. To answer this question, we chose to implement NMR measurements inside the transportation device. This will be the subject of a future study.

46) line 334, specify experimental conditions

We specified the conditions as in the following text: “To provide conditions for longer storage and/or transport, a colder environment is needed (i.e. below 4.2 K). At liquid He temperature, hours long T_1 can be obtained”

47) line 373, redundant sentence

OK. We removed the sentence. Now the paragraph writes:

“Finally, we want to stress that even in the ideal case of complete absence of paramagnetic impurities in the sample, for a magnetic field value $< 40\text{ mT}$ ^{13}C and ^1H nuclei are subjected to “low-field thermal mixing”.^{22,27}”

REVIEWERS' COMMENTS:

Reviewer #3 (Remarks to the Author):

Many thanks for addressing the comments, and congratulations to your great work.